# Training a Generally Curious Agent

**Fahim Tajwar** [* 1]   **Yiding Jiang** [* 1]   **Abitha Thankaraj** [1]   **Sumaita Sadia Rahman** [2]   **J Zico Kolter** [1]   **Jeff Schneider** [1]
**Russ Salakhutdinov** [1]

## Abstract

Efficient exploration is essential for intelligent systems interacting with their environment, but existing language models often fall short in scenarios that require strategic information gathering. In this paper, we present PAPRIKA, a fine-tuning approach that enables language models to develop general decision-making capabilities that are not confined to particular environments. By training on synthetic interaction data from different tasks that require diverse strategies, PAPRIKA teaches models to explore and adapt their behavior on a new task based on environment feedback in-context without more gradient updates. Experimental results show that models fine-tuned with PAPRIKA can effectively transfer their learned decision-making capabilities to entirely unseen tasks without additional training. Unlike traditional training, our approach's primary bottleneck lies in sampling useful interaction data instead of model updates. To improve sample efficiency, we propose a curriculum learning strategy that prioritizes sampling trajectories from tasks with high learning potential. These results suggest a promising path towards AI systems that can autonomously solve novel sequential decision-making problems that require interactions with the external world.

## 1. Introduction

Large language models (LLMs) are considered to be a promising foundation for autonomous agents, systems capable of achieving goals independently with minimal human supervision or intervention. A crucial requirement for such systems is the ability to interact effectively with external environments and gather the information necessary to achieve their objectives. This capability can be formalized as solving

*Equal contribution [1]CMU [2]North Carolina State University. Correspondence to: Fahim Tajwar <ftajwar@cs.cmu.edu>.

*Proceedings of the 42nd International Conference on Machine Learning*, Vancouver, Canada. PMLR 267, 2025. Copyright 2025 by the author(s).

sequential decision-making problems or performing reinforcement learning (RL) with language models as the agent. However, two challenges hinder the development of these interactive capabilities. First, most naturally occurring data lacks the structure and context needed to model interactions. Second, directly deploying models into the real world to collect interaction data can produce critical errors, which is expensive and potentially risky.

Given the impracticality of direct deployment in the wild, a natural alternative is to generate interaction data synthetically. Although generating synthetic data for every possible problem is infeasible, LLMs possess the capacity for *in-context learning* (ICL), which allows them to adapt to new tasks with minimal demonstrations (Brown et al., 2020). Instead of teaching the model to do all the interaction tasks that we care about, we should instead teach the model *in-context reinforcement learning* (Laskin et al., 2022) so that the model can solve new problems without being trained on them a priori. It shifts the focus from training the model on particular problems to training it on the general process of solving problems. This paradigm shares similarities with the supervised fine-tuning (SFT) and reinforcement learning from human feedback (RLHF) stages of training a language model (vs pretraining) where only a relatively small number of examples is needed to produce a model that can generate responses to a wide range of queries that they are not trained on. Our approach is also closely related to the principles of *meta reinforcement learning* (Beck et al., 2023).

In this work, we explore the feasibility of teaching LLMs to perform in-context RL that generalizes across different tasks, with the specific goal of training a curious agent with general information gathering capability. A popular notion of curiosity is *intrinsic motivation* which has been used to train agents with an exploration bonus not necessarily related to the success of any particular task (Schmidhuber, 1991; 2007). Our work differs from this notion of curiosity in that we do not leverage intrinsic motivation. Instead, we train our agents to explore and interact with an entirely unseen environment to gather information that is needed for completing the task at hand. PAPRIKA can be thought of as a form of *amortized exploration*, since our goal is to learn good exploration strategies from trajectories from many different environments to make exploration on a new

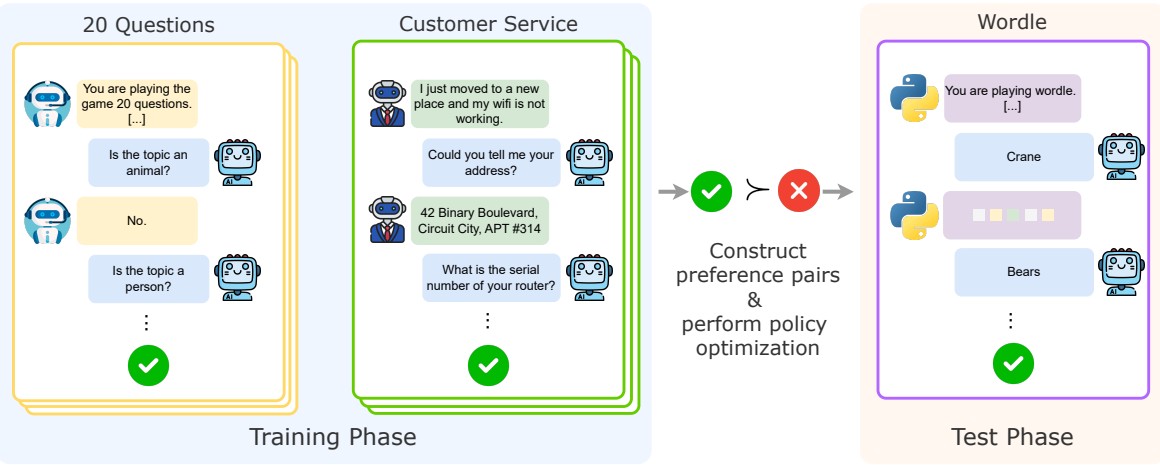

*Figure 1.* (**Overview of PAPRIKA**) We design a diverse set of tasks where an LLM agent needs strategic information gathering to succeed, then train an LLM on self-generated data to prefer higher performing trajectories. The resulting behavior learned by PAPRIKA can transfer zero-shot to unseen tasks, showcasing its potential to build general decision making agents.

problem more efficient (see Appendix A for more details).

We begin by designing a diverse suite of textual decision-making tasks that require active information gathering and decision-making based on interaction outcomes. Using a base model, we generate interaction trajectories and assign scores based on their success in achieving the tasks' objectives. We then apply a sequential variant of Direct Preference Optimization (Rafailov et al., 2024b, DPO) to increase the relative likelihood of successful trajectories. Unlike traditional training where computational costs are dominated by model updates, our approach's primary bottleneck lies in sampling useful interaction data. To improve sample efficiency, we propose a curriculum learning strategy that prioritizes sampling trajectories from tasks with high learning potential.

We refer to the overall framework as PAPRIKA[1]. Our results demonstrate that training on different subsets of these tasks improves the performance of the model on unseen tasks. More broadly, our result highlights the potential of using synthetic data to learn in-context RL which would equip LLMs with the capability to interact with the world and solve different decision-making problems without requiring task-specific fine-tuning.

## 2. Preliminary

Many decision making problems can be formalized as a partially observable Markov decision process (POMDP). We assume each *task*, $\tau$, is a POMDP although we will not draw on the details of the POMDP formalism in this work.

[1] The name is inspired by the movie "Paprika" (2006), where a dream detective navigates vast and strange dream worlds to solve different mysteries.

As a concrete example, guessing the word "apple" would be a task in 20 questions. We will use *group* (or *task group*, used interchangeably), $G = \{\tau_1, \tau_2, \ldots, \tau_{|G|}\}$, to refer to a high-level grouping of different tasks (e.g., the game 20 questions would be a group). Tasks in a group should share similar strategies but it is not always true that they share the same optimal policy as such constraints may be overly stringent. From the agent's perspective, each task is a black box function that takes in the agent's action $a_t$ (and possibly the whole interaction history) and outputs an observation $o_t$. Both $a_t$ and $o_t$ are strings. In a game of 20 questions, $a_t$ could be "Is the word an animal?" and the $o_t$ could be "No.". In other words, each task employs an environment that the agent interacts with to obtain intermediate observations.

An episode contains the agent's interaction trajectory within a single task. Unlike the conventional RL structure, we will assume that the transition-level reward is either 0 or must be inferred from $o_t$, and that the individual tasks can flexibly implement different observation spaces and termination conditions. An episode terminates when the agent achieves the objective of the task or when the maximum number of interactions allowed within the task is reached. We will use $h = (o_0, a_0, \ldots, o_H, a_H)$ to denote an episode of length $H$, $h_t = (o_t, a_t)$ to denote a single step of $h$, and $h_{p:q} = (o_p, a_p, \ldots, o_q, a_q)$ to denote a slice of $h$ similar to array slicing. At the end of an episode, the environment emits a single score, $r(h)$, that evaluates the performance of the agent. Let $\pi$ denote the LLM agent and $h \sim \pi \circ \tau$ denote sampling a trajectory from task $\tau$ using policy $\pi$. The performance of a policy on a group would be: $\text{Perf}(G) = \frac{1}{|G|} \sum_{\tau \in G} \mathbb{E}_{h \sim \pi \circ \tau}[r(h)]$. The agent is trained on a finite set of groups, $\mathcal{G}_{\text{train}}$, and the goal is to perform well on unseen groups, $\mathcal{G}_{\text{test}}$.

*Table 1.* Summary of the task groups used by PAPRIKA.

| Task Group | # Train Tasks | # Test Tasks | Maximum Turns | Env Feedback | Uses COT |
|---|---|---|---|---|---|
| Twenty questions | 1499 | 367 | 20 | LLM generated | ✗ |
| Guess my city | 500 | 185 | 20 | LLM generated | ✗ |
| Wordle | 1515 | 800 | 6 | Hardcoded program | ✓ |
| Cellular automata | 1000 | 500 | 6 | Hardcoded program | ✓ |
| Customer service | 628 | 200 | 20 | LLM generated | ✗ |
| Murder mystery | 203 | 50 | 20 | LLM generated | ✗ |
| Mastermind | 1000 | 500 | 12 | Hardcoded program | ✓ |
| Battleship | 1000 | 200 | 20 | Hardcoded program | ✓ |
| Minesweeper | 1000 | 200 | 20 | Hardcoded program | ✓ |
| Bandit best arm selection | 81 | 1 | 21 | Hardcoded program | ✓ |

## 3. PAPRIKA

The goal of our paper is to develop a scalable method to instill better strategic exploration and sequential decision-making capabilities into LLMs. Prior works (Krishnamurthy et al., 2024) have shown that LLMs can perform poorly on even the simple decision making task of multi-armed bandits. Nie et al. (2024) has since then demonstrated that LLMs can be taught to perform better on bandits after fine-tuning them on synthetic trajectories generated by known algorithms such as UCB. However, this idea is limited in scope for three reasons: **(1)** we want LLMs to perform strategic exploration and decision making in more complex settings, **(2)** for most tasks, there is no known algorithm like UCB to generate good synthetic trajectories from, **(3)** it can be infeasible to collect data for all tasks that we care about.

We aim to solve these issues using our method, PAPRIKA. First, we design a suite of complex decision-making tasks that require strategic information gathering to succeed. Next, we show that in the absence of known good algorithms, existing LLMs can generate trajectories with better decision making behaviors through diversity-encouraging sampling. We then finetune the LLMs to prefer higher performing trajectories (in a fashion similar to STaR (Zelikman et al., 2022)) and show that this leads to better decision making abilities at test-time. More importantly, these behaviors often generalize to unseen task groups without additional training. Finally, we propose a general curriculum learning algorithm that can dynamically choose which subset of tasks to train on next to improve data efficiency of such training methods. We next describe each component of PAPRIKA.

### 3.1. Task Design

The first component of PAPRIKA is to design a set of task groups that we can evaluate and train LLMs on. The task groups we want should have the following desired properties: **(1)** they are purely text based, **(2)** they require multi-turn interaction, where the agents have to both understand prior history in its context and choose actions that maximize the probability of success in the future, **(3)** they are partially observable, i.e., the observations do not capture the full state or hidden information, so the agents must simultaneously explore to reveal more information and exploit to solve the task efficiently, **(4)** they are diverse and require different strategies to succeed.

With these requirements in mind, we design 10 task groups in our paper. On all of them, we employ an LLM as the agent that is given a task it needs to solve through sequential interaction with the task-specific environment, which provides both observations for intermediate timesteps given the agent's actions and also a task reward at the end of an episode. For tasks requiring general knowledge about the world to generate intermediate observations, we employ another LLM (typically GPT-4o-mini) as the environment. For tasks that have rule-based observations and rewards, we find that using hardcoded programs as the verifier/observation generator is more reliable than LLMs, similar to DeepSeek-AI et al. (2025). In order to prevent reward hacking, we also use either another LLM or a hardcoded program as a judge to filter out unsuccessful trajectories that got incorrectly labeled as successful by the task environment (see Appendix D for more on environment hacking). We also find that for task groups requiring complex reasoning, letting the agent think using chain-of-thought (COT) prompting (Wei et al., 2022; Kojima et al., 2022) before generating a final answer improves its performance significantly, similar to ReAct (Yao et al., 2023). We provide a brief description of our task groups here, please refer to Table 1 for their summary and Appendix B for more details.

Following prior work (Abdulhai et al., 2023), we include classic guessing games like *twenty questions* and *guess my city* in our list of task groups. They require guessing a secret topic as quickly as possible by asking a sequence of questions and observing the answers. We also employ *Wordle* and *Mastermind*, where the agent needs to guess a secret 5-letter word and 4-digit code respectively. The environments

for these task groups provide feedback in terms of similarity between the guess and the target word/code, and the agent needs to refine their guesses in future turns to maximize information gathering. We design *customer service* and *murder mystery* as dynamic text-based task groups: an LLM plays the role of the task environment, which is provided with the criterion for task success and generates dynamic intermediate observations based on this criterion.

A desirable capability in LLMs is to code and refine based on interpreter feedback. To simulate this process with a toy case, we design *Cellular Automata*, where the agent needs to make inferences about the transition rule in 1D elementary cellular automata (Wolfram, 1983; Cook et al., 2004) by observing inputs and outputs. The agent receives the outputs generated from their predicted transition rule and they have to refine their predictions based on it. Next, we incorporate *Minesweeper* and *Battleship* based on classical games, which require the agent to interact with 2D grids to find hidden items within a fixed number of turns and refine their guesses based on per-turn observations.

Finally, we incorporate a modified version of the multi-armed bandit (Slivkins, 2024) task group from prior works (Krishnamurthy et al., 2024; Nie et al., 2024) with the following distinctions: **(1)** we let the agent employ chain-of-thought reasoning before choosing arms so that they can transfer good strategies learned from other tasks, **(2)** we let the agent interact with the task environment in a multiturn way, **(3)** instead of reducing regret, we work on the bandit best arm selection (Audibert & Bubeck, 2010; Wang et al., 2024a) problem, where we let the agent choose arms and observe rewards for a fixed number of turns and then measure its accuracy in deciding the arm with the highest reward. This is done to reduce computational cost over generating COTs for a large number of turns, since the difference in regret between different models is not meaningful when the number of turns is not large enough.

### 3.2. Dataset construction

In order to learn from these task groups, we must first generate data from them. It is crucial that the data we generate are diverse which would allow the model to learn different strategies without the risk of overfitting. We accomplish this by generating a large number of trajectories at a high temperature with Min-p sampling (Nguyen et al., 2024). Min-p sampling works by using an adaptive threshold $p_{\text{scaled}} \propto p_{\max}$, where $p_{\max}$ is the highest probability predicted by the model on the next token, to truncate the vocabulary to tokens that have a probability larger than $p_{\text{scaled}}$ and sample from them — this enables us to generate diverse yet coherent trajectories at a higher temperature. We note that training data generation for PAPRIKA could be improved by adopting more advanced methods for guiding

exploration such as Murty et al. (2024); Yang et al. (2024); however, we opt for sampling with high temperature for its simplicity and leave these other options for future work.

For each task in a set of chosen tasks (e.g., uniformly sampled), we generate $n_{\text{sample}}$ trajectories and then construct a preference pair $(h_w, h_l)$ where $h_w$ is the highest scoring trajectory (trajectory that succeeds and does so at the fewest number of turns) and $h_l$ is randomly sampled from the lower scoring (failed or takes substantially more turns to succeed) trajectories. We choose $h_l$ randomly instead of choosing the worst one to increase the diversity of our dataset. We treat $h_w$ and $h_l$ as proxies for desirable and undesirable behaviors. A dataset $\mathcal{D} = \left\{ \left( h^w, h^l \right)^{(i)} \right\}_{i=1}^N$ is a collection of such trajectory pairs.

### 3.3. Optimization

**Supervised fine-tuning.** If we take the winning episodes as the expert behavior, then we can discard the losing episode and maximize the likelihood of winning episodes:

$$\mathcal{L}_{\text{SFT}}(\mathcal{D}_{\text{SFT}}) = -\mathbb{E}_{\mathcal{D}_{\text{SFT}}} \left[ \frac{1}{\sum_{t=0}^{|h_w|} |a_t^w|} \sum_{t=0}^{|h_w|} \log \pi_\theta \left( a_t^w \mid h_{:t}^w \right) \right] \tag{1}$$

where $\mathcal{D}_{\text{SFT}}$ is the dataset used for supervised fine-tuning and $|a|$ is the number of tokens for the agent response (discarding the environment generation). This is akin to rejection sampling fine-tuning (Gulcehre et al., 2023; Dong et al., 2023; Mukobi et al., 2023) seen in prior work.

**Direct preference optimization.** A popular approach for finetuning LLMs is DPO (Rafailov et al., 2024b) where one directly optimizes the Bradley-Terry model (Bradley & Terry, 1952) for preferences. In our setting, each trajectory consists of multiple rounds of interactions so the original DPO objective does not apply. We instead use a multi-turn version of DPO introduced in Rafailov et al. (2024a):

$$\mathcal{L}_{\text{DPO}}(\mathcal{D}_{\text{DPO}}) = -\mathbb{E}_{\mathcal{D}_{\text{DPO}}} \left[ \log \sigma \left( \sum_{t=0}^{|h^w|} \beta \log \frac{\pi_\theta(a_t^w \mid h_{:t}^w)}{\pi_{\text{ref}}(a_t^w \mid h_{:t}^w)} \right. \right.$$
$$\left. \left. - \sum_{t=0}^{|h^l|} \beta \log \frac{\pi_\theta(a_t^l \mid h_{:t}^l)}{\pi_{\text{ref}}(a_t^l \mid h_{:t}^l)} \right) \right] \tag{2}$$

where $\mathcal{D}_{\text{DPO}}$ is the preference dataset, $a_t^w$ and $a_t^l$ are the action tokens generated by the model at turn $t$ in the preferred and dispreferred trajectories, $h^w$ and $h^l$, respectively. $\pi_{\text{ref}}$ is the reference policy, for which we use the initial model. The main difference with standard DPO here is that we only

calculate the loss on the action tokens — the log probability ratios of the environment generated tokens are not included in the loss.

We note that we use DPO because it is less compute intensive. DPO allows us to decouple the data collection and policy improvement steps and offload them on different machines. However, in principle, one could also employ online RL with more resources. Following prior work that shows the efficacy of online RL compared to offline algorithms (Xu et al., 2024; Tajwar et al., 2024), we expect doing PAPRIKA with online RL would lead to even stronger results.

**Combining objectives.** Finally, prior works have noted DPO having the unintended effect of reducing the probability of preferred trajectories as well, known as unintentional unalignment (Razin et al., 2024), which can affect model performance. The RPO objective (Pang et al., 2024), by combining SFT and DPO loss, has shown promising results in mitigating this issue. Formally, the RPO loss is:

$$\mathcal{L}_{\text{RPO}}(\mathcal{D}_{\text{DPO}}) = \mathcal{L}_{\text{DPO}}(\mathcal{D}_{\text{DPO}}) + \alpha \mathcal{L}_{\text{SFT}}(\mathcal{D}_{\text{DPO}}) \quad (3)$$

where $\alpha$ is a hyper-parameter. Following Pang et al. (2024), we set $\alpha$ to be 1.0 for the rest of this paper.

### 3.4. Scalable Online Curriculum Learning

The core idea of PAPRIKA is to fine-tune the model on a large number of decision making problems to acquire general decision making ability. It is relatively easy to design a large number of tasks, but it is harder to decide which task to train on. A major obstacle is that different tasks may have a large range of difficulty. Unlike pretraining where the model can generally make progress on any given sample (i.e., decrease next-token prediction loss), an RL agent cannot make meaningful progress without collecting good experience. As such, if a task is too difficult for the current model, the model would not generate trajectories with meaningful learning signals. Since generating a trajectory is expensive, it stands to reason that we want to prioritize the tasks where the model can make meaningful progress, which is a form of curriculum learning (Bengio et al., 2009).

Without additional assumptions, the only way to know whether a task would yield good learning signals is to actually perform a rollout in that task, which is expensive. In fact, in this particular scenario, the major cost for training is actually data generation rather than model updates. As such, this naive approach would not save us time or computation. A desideratum for an efficient curriculum is the ability to know whether certain tasks will yield data with learning signals without actually performing the rollout. A natural assumption is that similar tasks would have similar levels of learning signal. These groupings can be obtained through

---

**Algorithm 1** Task selection with UCB

1: **Input:** Number of arms $K$, number of samples $C$, number of rounds $T$, model $\pi$
2: **Initialize:** $s_k = 0, n_k = 0$, Buffer
3: **for** each round $t = 1, 2, \ldots, T$ **do**
4:     Compute $\theta_k = \frac{s_k}{n_k} + \sqrt{\frac{2 \log \sum_{k=1}^{K} n_k}{n_k}}$ for each $k$
5:     Select $k^\star = \arg\max_k \theta_k$
6:     Sample $\tau$ from group $k^\star$
7:     Sample $C$ trajectories from $\tau$ and add to Buffer
8:     Compute an estimate for $\hat{\nu}_\pi(\tau)$ using Eq 4
9:     Update: $s_{k^\star} = s_{k^\star} + \hat{\nu}_\pi(\tau)$, $n_{k^\star} = n_{k^\star} + 1$
10: **end for**
11: Construct $\mathcal{D}$ from Buffer and train the model $\pi$

---

meta data or prior knowledge.[2]

**Measuring learning potential.** We will use $h \sim \pi \circ \tau$ to denote sampling one episode from the task $\tau$ using the policy $\pi$. The average performance of $\pi$ on $\tau$ is $R_\pi(\tau) = \mathbb{E}_{h \sim \pi \circ \tau}[r(h)]$ and the variance is $\sigma_\pi^2(\tau) = \mathbb{E}_{h \sim \pi \circ \tau}[(r(h) - R_\pi(\tau))^2]$. Based on these, we can define:

$$\nu_\pi(\tau) = \frac{\sqrt{\sigma_\pi^2(\tau)}}{R_\pi(\tau)}. \quad (4)$$

This quantity is known as the coefficient of variation in statistics, a dimensionless quantity that measures the population's variability relative to the mean.

We argue that this quantity is an ideal measure of the learning potential for a single task. DPO requires a pair of positive and negative samples [3]. Intuitively, the pair should be sufficiently different so the model can tell the two apart — for example, prior work (Pal et al., 2024) has shown that DPO suffers when the edit distance between preferred and dispreferred responses is not large enough. Variance naturally measures the possibility of getting diverse trajectories from sampling. On the other hand, different tasks could have vastly different reward scales. Without loss of generality, if we assume that all rewards are positive, the average reward of each task is a measurement of the reward scale. Normalizing the standard deviation with the reward scale allows us to compare different tasks directly.

**Sampling tasks.** Each group contains a large number of different tasks. Since it is infeasible to evaluate $\nu_\pi(\tau)$ for all tasks, we instead sample tasks from the group. This induces a scalar distribution that describes the distribution

---

[2]While this requirement may seem restrictive, we believe assumptions of similar effects are likely needed for any form of curriculum learning to be computationally efficient.

[3]We hypothesize this quantity would also apply to online RL since if all sampled trajectories have the same reward the policy gradient update would be 0.

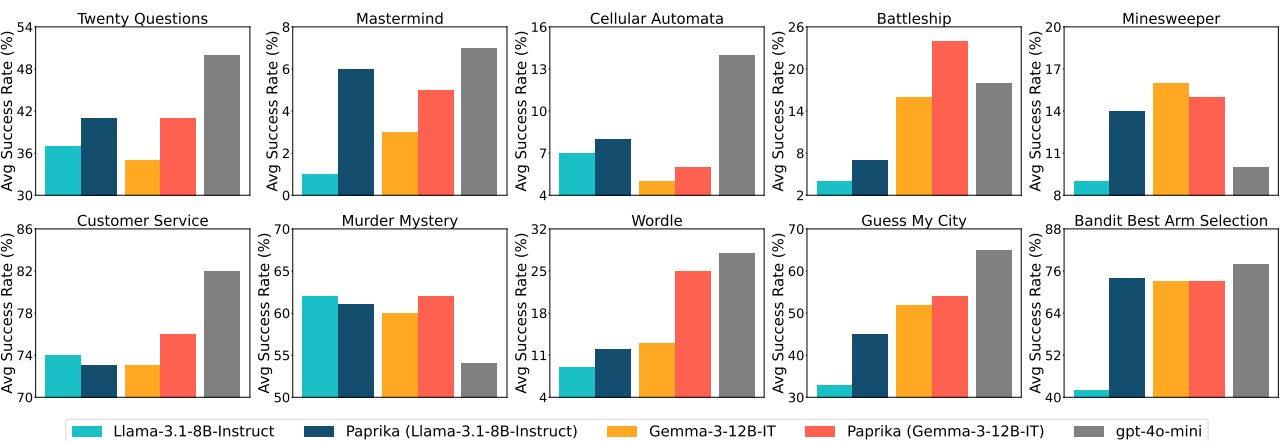

*Figure 2.* (**PAPRIKA improves success rate on a diverse range of task groups**) Average success rate on all 10 task groups at temperature 0.7. PAPRIKA generally improves performance of both Llama-3.1-8B-Instruct and Gemma-3-12B-IT models.

of $\nu_\pi(\tau)$ for all tasks in the group $G$. Given a collection of $K$ groups $(G_1, \ldots, G_K)$, a reasonable objective would be to maximize the learning potential of the tasks sampled. This problem can be formulated as a multi-armed bandit (MAB). Many algorithms for MAB exist; for simplicity, we choose the Upper Confidence Bound (Auer, 2000, UCB).

We conduct the task selection in a sequential manner using the original UCB algorithm, but we expect a batched variant of UCB could be used to parallelize the experience collection. Each action corresponds to a group of tasks, and we then uniformly sample one task from the chosen group to evaluate the model performance with $C$ rollouts. These statistics are then used to update the mean estimate of that group. After a sufficient amount of episodes are sampled, we construct the dataset and train the model with objectives in Section 3.3. See Algorithm 3.4 for the pseudocode.

**Note.** An important role of $\nu_\pi$ is to make different task groups comparable. The specific selection algorithms could likely be replaced with other more sophisticated online learning methods. More importantly, recent breakthroughs such as OpenAI et al. (2024b) and DeepSeek-AI et al. (2025) mark the beginning of applying RL to a broad range of reasoning problems. Moving forward, we anticipate a proliferation of different RL tasks for LLMs. In this emerging paradigm, a scalable meta algorithm for selecting which tasks to train on will be essential, and we believe PAPRIKA's curriculum learning approach will be a promising foundation for future algorithms.

## 4. Empirical Results

In this section, we will present the results of our empirical study to answer the following research questions: **(1)** Can training on self-generated trajectories from a diverse range of task groups equip LLMs with sequential decision making

capabilities that generalize to unseen task groups without the need to train on them? **(2)** Can curriculum learning improve the data efficiency of our training mechanism? **(3)** Finally, does PAPRIKA hurt the model's regular abilities, and can fine-tuning on existing multiturn interaction data that do not have any sequential decision making structure also improve these capabilities? We first describe our experimental setup, and then report our empirical observations.

**Experimental Setup.** For experiments in this paper, we use Llama-3.1-8B-Instruct (MetaAI et al., 2024) and Gemma-3-12B-IT (Gemma-Team et al., 2025) models. For data generation, we use Min-p sampling (Nguyen et al., 2024) with temperature 1.5 and Min-p parameter 0.3, as we saw that this setting consistently generated diverse training data that resulted in higher test-time accuracy. For each task in the training split, we generate $n_{\text{sample}} = 20$ trajectories to construct our training dataset (except for mastermind, where we sample $n_{\text{sample}} = 100$ trajectories per task). After filtering, this results in 17,181 training trajectories for supervised fine-tuning and 5,260 trajectory pairs for RPO over all task groups. Unless explicitly mentioned otherwise, we use learning rate of $10^{-6}$ for supervised fine-tuning and $2 \times 10^{-7}$ for RPO. We use batch size 32 for all training runs. We generally always run supervised fine-tuning first and then further fine-tune with the RPO objective to obtain the final model unless explicitly mentioned otherwise. We use an AdamW optimizer (Loshchilov & Hutter, 2019) with a cosine annealing learning rate scheduler and warmup ratio 0.04 (Loshchilov & Hutter, 2017) to train all our models.

During evaluation, in order to account for variability of both the environment and the agent, we generate 4 trajectories for each task in the test set and report the average success rate (we also report pass@4 success rates in Appendix I). We use Min-p sampling with parameter 0.3 for evaluation. Default temperature for evaluation is set to 0.7. Finally,

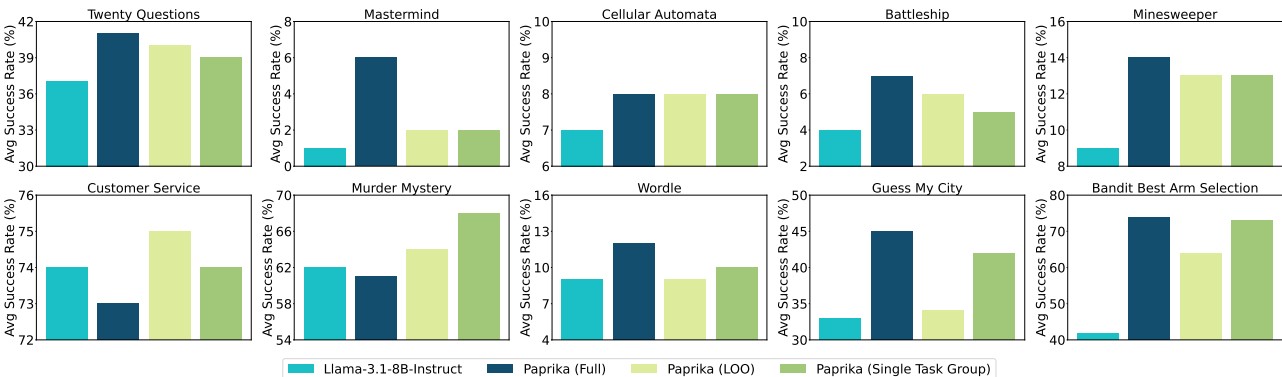

*Figure 3.* (**Testing generalization of PAPRIKA via leave-one-out and single task group experiments**) We test PAPRIKA's zero-shot performance on unseen task groups by leave-one-out (LOO) experiments, where we train the LLM on every task group except the group we test on. We also report the performance of PAPRIKA (Single Task Group), where we train and test the LLM on a single group. Our experiments demonstrate that PAPRIKA can teach an LLM decision making abilities that often transfer well to new tasks without any additional training, and the model also generally learns better in-group strategies when it observes trajectories from other task groups.

for task groups with hardcoded feedback mechanism, we consider a failure to follow formatting instructions to be a failure in the task.

**PAPRIKA improves LLM decision making abilties.** We motivate this question by looking into the toy task group of bandit best arm selection more closely. This task requires strategic use of the fixed sampling budget (20) to quickly discard arms that are unlikely to have a high mean reward, and use most of the sampling budget on the few top arms to decide the best arm among them. Previous work (Nie et al., 2024) has shown that training on synthetic trajectories from optimal bandit algorithms can significantly improve LLMs' performance on them. Contrary to that, we show that LLMs can learn generalizable strategies from other decision making task groups that then transfer to this bandit group, without needing an optimal algorithm to generate synthetic trajectories. Figure 3 shows that PAPRIKA improves average success rate of Llama-3.1-8B-Instruct from 42.25% to 62.25% on the bandit task after only seeing trajectories from other task groups.

Motivated by this, we next study whether PAPRIKA can also improve performance on more complex tasks. Figure 2 shows our main findings: PAPRIKA, when trained on a dataset consisting of filtered trajectories from all 10 task groups, improves the success rate of both Llama-3.1-8B-Instruct and Gemma-3-12B-It models (see Appendix I for complete results). Averaged across all 10 task groups, PAPRIKA increases the Llama-3.1-8B-Instruct model's performance by 47% of its original success rate after training with only about 22,500 trajectories.

**PAPRIKA can teach LLMs generalizable strategies.** The next important question we want to study is whether the strategies learned by PAPRIKA can zero-shot transfer

to entirely different groups of tasks. We saw already that PAPRIKA (LOO) improved the success rate on the bandit group without the need to train on it, now we explore this possibility for more complex decision making tasks. To do so, we perform a set of leave-one-out (LOO) experiments: we randomly choose one group (e.g., 20 questions) from our set of task groups, train the LLM on trajectories generated from every other group, and test the resulting model's performance on the left-out group. Additionally, we run an experiment where for each task group, we train and test the LLM on only this single group (using separate splits). We use Llama-3.1-8B-Instruct for this set of experiments.

Figure 3 shows our results: remarkably, we observe that the LOO models can match or sometimes even exceed the performance of group-specific training, demonstrating genuine cross-task group generalization. Concretely, PAPRIKA (LOO) improves success rate on 9 out of 10 task groups compared to the initial model. Moreover, PAPRIKA (full), trained on all 10 task groups, outperform PAPRIKA (Single Task Group) in 7 out of 10 task groups, showing that the model learns better in-group strategies when it observes trajectories from other task groups. Note that we do not expect PAPRIKA (LOO) to always generalize to a new task group. While PAPRIKA (LOO) generalizes better to some task groups vs others (e.g., the improvement on mastermind is minimal), and for some task groups there is no transfer at all or negative transfer (wordle), we hypothesize that scaling up the number of task groups could keep improving LLMs' zero-shot decision-making abilities. Overall, these results demonstrate that PAPRIKA is a potentially scalable solution for teaching LLMs how to do in-context RL.

**Curriculum learning can improve data efficiency of PA-PRIKA.** The biggest bottleneck of PAPRIKA is the time required to generate a large number of trajectories for each.

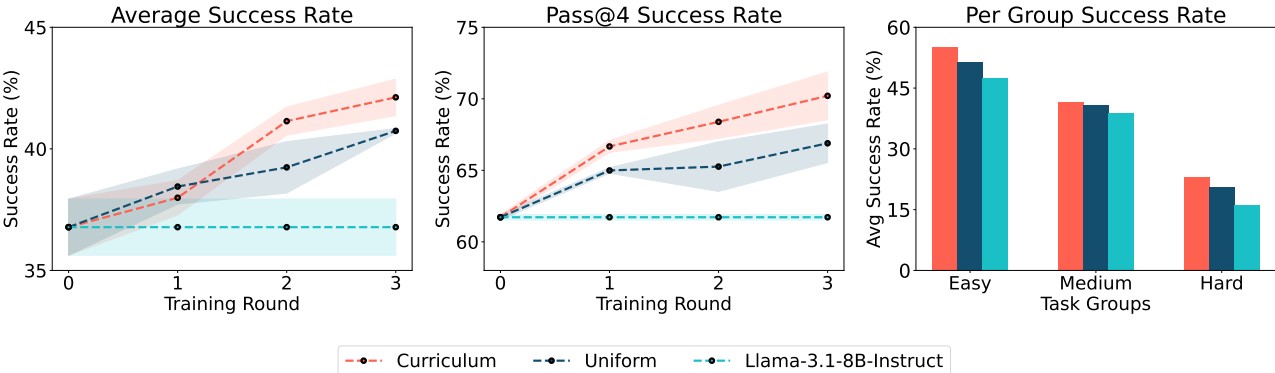

*Figure 4.* (**Multi-round training with curriculum on twenty questions**) We demonstrate the efficacy of our curriculum learning algorithm for sampling training tasks by comparing its performance against uniform sampling for multi-round training. All experiments use Llama-3.1-8B-Instruct as the initial model, evaluations are done at temperature 0.7, and shaded regions represent standard error over 3 seeds. (**Left**) Average success rate at each round. (**Middle**) Pass@4 success rate at each round. (**Right**) Success rate per each of easy, medium, and hard task groups. Overall, our curriculum learning algorithm shows 1.4% and 3.3% improvement over the uniform sampling baseline at average and pass@4 success rate respectively.

Some tasks are naturally harder than others, which means that spending an equal sampling budget on the harder tasks gives us a smaller learning signal. We study a curriculum learning version of PAPRIKA where we have a grouping over our tasks according to task difficulty. For this, we use GPT-4o-mini to classify the tasks in twenty questions into 3 categories: easy, medium, and hard. This results in 477 easy, 726 medium, and 296 hard topics in the train split and 127 easy, 172 medium, and 68 hard topics in the test split.

Next, we run the curriculum learning algorithm described in Section 3.4 for 3 rounds on a Llama-3.1-8B-Instruct model: at each round, we sample 250 tasks from the train set according to Section 3.4. We use the number of turns it took the agent to solve a task across multiple trajectories as a proxy for reward in Equation (4) to calculate $\nu_\pi$ (see Appendix H for more details). 20 trajectories are generated for each task using the previous round's model checkpoint and we train that checkpoint on the resulting dataset (for DPO, we use the prior round's checkpoint instead of the initial model as the reference policy). We compare our curriculum against the baseline of sampling 250 tasks uniformly at random from the train set at each round. Figure 4 shows our results: after three rounds of training, our curriculum outperforms uniform sampling by 1.4% and 3.3% at average and pass@4 accuracy respectively.

### 4.1. Analysis

**PAPRIKA improves LLMs' task efficiency.** In this section, we want to analyze the sequential decision-making abilities learned by PAPRIKA beyond just success rate on individual task groups. Note that our tasks are designed in a way such that an agent capable of better strategic exploration would solve them faster, eg., an agent capable of asking better yes/no questions would guess the secret topic

using fewer number of turns. We leverage this property of our tasks and conduct both quantitative and qualitative analysis on the behaviors of the regular instruct model and PAPRIKA — (**1**) Figure 7 shows that PAPRIKA reduces the average number of turns it takes for the agent to solve tasks, implying that PAPRIKA is choosing more optimal actions at intermediate steps, (**2**) Appendix K shows qualitative difference between the behavior of the regular instruct model and PAPRIKA on twenty questions and wordle, with PAPRIKA generally generating more sensible responses.

**PAPRIKA does not hurt LLMs' regular capabilities.** We have demonstrated the efficacy of PAPRIKA in instilling decision making capabilities into LLMs efficiently. However, to scale up PAPRIKA, one would potentially use online reinforcement learning on such decision making tasks, and an important question is whether PAPRIKA fine-tuning would hurt the LLM's regular capabilities which would hinder scaling it up. To study this question, we run a set of standard evaluations (see Appendix I.12) on our PAPRIKA fine-tuned model and compare its performance against Llama-3.1-8B-Instruct. Table 2 shows our findings: PAPRIKA does not result in any noticeable performance degradation.

## 5. Related Works

**LLM alignment.** Alignment or post-training is a crucial step for creating helpful LLM assistant. Existing post-training pipeline typically involves instruction tuning and then reinforcement learning from human feedback (Christiano et al., 2017, RLHF) where one either performs RL against a reward model trained on human preference data via Proximal Policy Optimization (Schulman et al., 2017, PPO) or sidesteps reward model training via Direct Preference Optimization (Rafailov et al., 2024b, DPO). Most methods

*Table 2.* (**Evaluation of PAPRIKA on standard tasks**) Evaluation of PAPRIKA vs Llama-3.1-8B-Instruct on standard benchmarks (numbers in parenthesis represent standard error over 3 seeds). PAPRIKA does not result in significant model degradation.

| Model | MT-Bench | AlpacaEval | GPQA | Math (Hard) | MMLU-Pro | IFEval |
|---|---|---|---|---|---|---|
| Llama-3.1-8B-Instruct | 7.88 | **33.6** | **33.5** | 24.6 | **46.7** | 84.4 |
| + PAPRIKA | **8.14 (0.03)** | 33.5 (0.3) | 32.8 (1.5) | **25.3 (0.3)** | 46.2 (0.1) | **85.4 (0.3)** |

focus on *single-turn* interactions where the model generates a single response to a query. We focus on the *multi-turn* setting where the agent has to interact with an environment iteratively, similar to Rafailov et al. (2024a). There are a few existing environments and datasets that focus on multi-turn interactions (Abdulhai et al., 2023; Sun et al., 2023; Kwan et al., 2024; Wang et al., 2024b). LMRL-Gym (Abdulhai et al., 2023) implements a suite of textual RL environment, some of which we build on. Concurrent work such as Narayanan et al. (2024) has designed environments based on scientific tasks (such as molecule cloning and protein stability) for LLMs to interact with and showed that behavior cloning and expert iteration (Anthony et al., 2017; 2019; Havrilla et al., 2024) can improve an LLM's multi-turn interaction capabilities on these tasks. Most of these environments focus on interactions with humans. Rather than any particular task, we focus on evaluating LLMs' general ability to solve sequential decision making problems where the agent needs to explore and exploit.

**In-context reinforcement learning.** In-context learning (ICL) is the ability where LLMs can learn a new task from a small number of demonstrations without any gradient update (Brown et al., 2020). Existing ICL usually focuses on a single-turn interaction. We focus on in-context reinforcement learning (Laskin et al., 2022; Raparthy et al., 2023; Lee et al., 2024; Lin et al., 2024) instead. Existing work in this field has focused on environments where RL is conventionally applied (e.g., grid world, bandits, and maze) (Monea et al., 2025), and the training data are generated by either random policies or pre-existing RL algorithms. In comparison, we focus on diverse environments and study how well the decision making abilities generalize to completely new environments. Concurrent work has also studied improving LLMs' information seeking abilities (Li et al., 2025) for medical reasoning, whereas we work on general information seeking abilities applicable to a diverse range of tasks. Moreover, Harris & Slivkins (2025) has studied using an LLM to assist a decision-making agent navigate exploration-exploitation tradeoff, whereas we use an LLM directly as the decision making agent and teach it this capability.

**Curriculum learning in RL.** Curriculum learning (Bengio et al., 2009) shows the data to the model in a non-uniform order. This idea is inspired by the fact that humans tend to learn skills in a sequential order (Skinner, 1958),

and is particularly appealing for RL because learning easier tasks first could build scaffold toward solving difficult tasks that the agent could not solve otherwise (Andrychowicz et al., 2017; Florensa et al., 2017; Fang et al., 2019; Portelas et al., 2020a). Concurrent work such as Foster & Foerster (2025) has studied curriculum learning for training LLMs to improve their reasoning capabilities. While their work requires generating rollouts per each example to determine the learnability, we show that given access to some grouping metadata, one can design an effective curriculum using only a constant number of rollouts generated from each task group. Another related line of work is environment design, where a second process controls the distribution over different environments or directly generates environments in a procedural manner to maximize various notions of learning progress (Wang et al., 2019; Dennis et al., 2020; Jiang et al., 2021b;a; Bruce et al., 2024). Since this is a field of extensive existing literature, we refer the interested reader to Portelas et al. (2020b) for a comprehensive survey.

## 6. Discussion

In this paper, we presented a scalable fine-tuning method to improve multi-turn decision making abilities of LLMs. Moreover, we showed that the strategies learned by the LLM from our method can generalize zero-shot to unseen tasks. There are a few limitations to our approach. Firstly, we use rejection sampling on self-generated data to teach the model better behaviors. In order to get good performance, the starting model need to exhibit good behavior within a reasonable generation budget, so PAPRIKA would perform worse in the absence of a good base model. Next, we use offline preference tuning algorithms to train our models due to the lack of computational resources. A possible future direction for our work is to run online RL on diverse tasks instead: due to its recent success in other domains (DeepSeek-AI et al., 2025), we expect it will give a larger improvement in LLMs' in-context RL capabilities. Our environments, despite being designed with the help of GPT-4o-mini, required a lot of human effort for implementation. A new axis of improvement can be training an LLM to scalably generate suitable tasks that can then be used to train the agent. Finally, the performance of our curriculum learning algorithm heavily depends on the quality of the task group clusters which is not ideal, and one can study possible improvements of this algorithm. We leave these directions for future work.

## Impact Statement

Our work can be used to train large language models that have better strategic exploration and decision making capabilities, which can have potential impact in the real world if agentic systems become wide spread. Our experiments are conducted in relatively simple and controlled environments and it is an open question what kind of impacts truly agentic systems will have on society. Other than that, this paper presents work whose goal is to advance the field of Machine Learning. There are many potential overall societal consequences of our work, none of which we feel must be specifically highlighted here.

## Reproducibility Statement

We provide sufficient details about our implementation, hyperparameters, environment design and dataset construction in the main paper and the appendix to effectively reproduce the results in this paper. Our code, training dataset and models can be found via the project website: https://paprika-llm.github.io/

### Acknowledgement

This work was supported in part by the U.S. Army Futures Command under Contract No. W519TC-23-C-0030. Moreover, it has greatly benefited from using the Delta advanced computing and data resource supported by the National Science Foundation (OAC 2005572) and the State of Illinois, as part of ACCESS-approved compute grants (Boerner et al., 2023). Subsequent larger scale experiments on Gemma-3-12B-IT models were run using Bridges-2 (Brown et al., 2021) at Pittsburgh Supercomputing Center through ACCESS allocation CIS240901 from the Advanced Cyberinfrastructure Coordination Ecosystem: Services & Support (ACCESS) program, which is supported by National Science Foundation grants #2138259, #2138286, #2138307, #2137603, and #2138296. The authors thank Brandon Pusateri, Jillian Lehosky and Greg Bauer from ACCESS Support Staff for their incredible help at approving supplements and renewals for ACCESS compute grants throughout this project. Moreover, the work would not have finished so quickly without the help of Brett Bode from NCSA Delta Support Staff, who provided the authors critical help about properly utilizing the Delta cluster. FT and YJ gratefully acknowledge Samuel Sokota, Daman Arora, Andrea Zanette, Yuda Song, Gaurav Ghosal, Yutong He, So Yeon Min, Kevin Li, Wen-Tse Chen, Xintong Duan and other members of Russ, Auton, Locus and AIRe lab for feedback received on an earlier versions of this work. FT greatly benefited from his discussions with Prof. Aviral Kumar and his lab's computational resources. YJ gratefully acknowledges the support of the Google PhD Fellowship.

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

# A. Note on Curiosity

The concept of curiosity has been used in many different machine learning contexts. A popular notion of curiosity is *intrinsic motivation*, where the agent is driven by an exploration bonus that is not necessarily related to the task to be achieved (Schmidhuber, 1991; 2007). Many works build on this notion to handle problems with sparse reward or no reward at all (Pathak et al., 2017; Eysenbach et al., 2018; Burda et al., 2018; Sharma et al., 2019; Pathak et al., 2019). The curiosity in this work differs from intrinsic motivation in that we focus on gathering only the information required to solve a given task rather than all the knowable information. This is closer in spirit to the original exploration-exploitation trade-off in reinforcement learning (Sutton et al., 1998; Auer et al., 2002; Thompson, 1933). The goal is to explore to the extent that the problem can be solved but not over-explore at the cost of efficiency. Most existing works based on this principle are *tabula rasa* (Osband et al., 2016; Chen et al., 2017). This class of exploration algorithms has been shown to improve the generalization ability of non-LLM-based RL agents (Jiang et al., 2023b). PAPRIKA differs from these approaches by learning good exploration strategies from trajectories from many different environments to make exploration on a new problem more efficient. This can be thought of as a form of *amortized exploration*.

# B. Details on Task Design

## B.1. Summary of Task Groups

**Twenty questions:** Twenty questions challenges the agent to identify a secret topic by asking up to 20 yes-or-no questions. The goal is to guess the topic in as few questions as possible by interpreting previous answers and strategizing to maximize information gained. Twenty questions has been studied in prior benchmarks such as LMRL-Gym (Abdulhai et al., 2023): here we expand upon their environment with a more diverse and difficult set of secret topics. Our secret topics come from a diverse range of scenarios, including famous people, historical events, scientific concepts, locations, etc. Each secret topic corresponds to a task, and we have generated a set of 1499 train and 367 test tasks. In order to generate a diverse set of topics, we use prompting techniques from GenQA (Chen et al., 2024) on GPT-4o-mini. The topics to guess in our training and test sets are distinct from one another and also the set of topics included in LMRL-Gym (159 topics), which use as an additional evaluation set. We use GPT-4o-mini (Hurst et al., 2024; OpenAI et al., 2024a) as the task environment to provide yes/no answers at every turn, and also as a judge to make sure task success label is correct. We use strict string matching to make sure the intermediate observations are only 'yes', 'no' or 'Goal reached'. We also maintain train and test set separation to accurately test generalization unlike previous works.

**Guess my city:** Following LMRL-Gym, this task group requires the agent to guess a secret city after asking a maximum of 20 questions. But unlike twenty questions, the questions here can be broader than just yes/no questions, for example, "*What is your city most popular for?*" so long as the answer to the question does not reveal the name of the city directly. We generated a train set of 500 and test set of 185 distinct cities using GPT-4o-mini and GenQA (Chen et al., 2024) prompting techniques. In addition, we also evaluated our models on the list of 91 cities from LMRL-Gym, which does not overlap with our training and test set. We maintain train and test set separation.

**Customer service:** In this task group, we test for efficient directed exploration —- the LLM must act as a support agent who asks maximally informative questions to diagnose problems and minimize the number of interactions needed to resolve the customer's query. To do so, we simulate realistic troubleshooting scenarios ranging from electronic device issues to automobile maintenance. We use GPT-4o-mini to simulate a customer with limited technical expertise, and use another LLM to act as a customer service agent whose role is to listen to the responses from the customer and suggest a sequence of actions that lead to solving the customer's problem in as few turns as possible. The customer service troubleshooting scenarios are generated by GPT-4o-mini, using prompting techniques from GenQA.

**Murder mystery:** Text-based interactive fiction (IF) environments can be a good benchmark to test LLMs' decision making and interaction abilities. Inspired by Hausknecht et al. (2020a), we design our murder mystery task group, where an LLM is given a crime scene with a possible list of suspects, witnesses, and clues, and it needs to take actions to uncover more information to successfully determine the culprit. The environments provided in Hausknecht et al. (2020a) proved difficult to incorporate directly in our setup, since they have a predefined list of valid actions and uses text-based parsing on the LLM generation to match against the list, making it difficult for LLMs to play the games. Instead, we use GPT-4o-mini to simulate the environment that can provide dynamic feedback to the agent's actions. The murder mystery scenarios are generated by GPT-4o-mini, using prompting techniques from GenQA.

**Wordle:**    Wordle tests an LLM's deductive reasoning abilities. The agent must guess a secret 5-letter word within 6 attempts. After each guess, the environment provides feedback for each letter: correct letter in correct position, correct letter in wrong position, or letter not in the word. The agent must use this feedback strategically to maximize information gained with each guess. We found that LLMs like GPT-4o-mini cannot generate accurate environment feedback for Wordle, so we use hardcoded rules to generate it instead. We also saw that prompting the LLM agent to do chain-of-thought reasoning before outputting its final guess significantly improves its performance, so we use that here unlike the environments above. The secret words are generated by looking at 5-letter words from an English dictionary.

**Cellular Automata:**    A key trait of LLM agents is the ability to code and refine based on interpreter feedback. To model this, we create a cellular automata-based environment. Here, a binary string (e.g., 1010) represents cells, and a transition rule defines a cell's next state based on itself and its neighbors (e.g., 100: 1 means a 0 cell with 1 and 0 neighbors turns into 1). We randomly select a transition rule (one of 256) and up to three input strings and their corresponding outputs generated by the transition rule. The LLM must infer the rule by analyzing input-output pairs. If its guess generates correct outputs, it wins; otherwise, it gets feedback and can refine its guess. The task ends in failure if the correct rule isn't found within six turns. We use chain-of-thought prompting for the agent and a hardcoded program to generate environment feedback. The tasks are generated by sampling transition rules and inputs randomly.

**Mastermind:**    Similar to Wordle, Mastermind challenges agents to deduce a 4-digit secret code within 12 turns. After each guess, environment feedback indicates two values: the number of digits that are correct and in the right position (exact matches), and the number of digits that appear in the code but in wrong positions (partial matches). Agents must use this feedback to iteratively refine subsequent guesses. We use chain-of-thought prompting for the agent and a hardcoded program to generate environment feedback. The tasks are generated by randomly sampling (without replacement) secret codes from all possible 10,000 four digit codes.

**Battleship:**    Battleship tests an LLM's ability to balance exploration and exploitation. The environment features a 2D square grid where three ships are hidden: a carrier (5 cells), a battleship (4 cells), and a destroyer (2 cells). Ships are placed horizontally or vertically. At each turn, the agent targets one cell with a missile. The environment environment reports either a hit (including the ship type) or a miss. A ship sinks when all its cells are hit. The agent must sink all ships within 20 turns. This environment environment requires grid exploration to locate ships and once located, exploitation in the form of targeted attacks to sink them. We use chain-of-thought prompting for the agent and a hardcoded program to generate environment feedback. The tasks are generated by randomly choosing the ship locations at each iteration.

**Minesweeper:**    We include minesweeper to test an LLM's sequential logical reasoning ability. The agent interacts with a 2D rectangular grid containing hidden mines. At each turn, the agent reveals one cell. The first move is always safe since mines are placed afterwards. If a mine is revealed, the task ends in failure. To win, the agent must reveal all mine-free cells within 20 turns. When a cell is revealed, it displays a number indicating how many mines are in adjacent cells. If a revealed cell has no adjacent mines (shown as '0'), all neighboring mine-free cells are automatically revealed. We use chain-of-thought prompting for the agent and a hardcoded program to generate environment feedback. The tasks are generated by randomly placing mines in the 2D grid at each generation.

**Bandit Best Arm Selection:**    Multi-arm bandits are a classic test for an agent's ability to perform sequential decision making — LLMs have been tested on this task in prior works such as Krishnamurthy et al. (2024); Nie et al. (2024). In this environment, an LLM is presented with a hypothetical scenario where it can select arms at every turn and observe the reward chosen from a Bernoulli distribution with a fixed but unknown mean attached to that arm. We created a modified version of their environment with three key distinctions: 1) prior works operated on bandits in a single-turn fashion: at each turn, LLMs were given the problem setup and history of past interactions within a single user prompt and asked to choose the next arm. Instead, our design employs multi-turn interactions, where the task description is given in the first turn, and later turns only provide rewards for the selected arm. 2) Prior works required the LLM to output only the chosen arm, whereas we employ chain-of-thought (COT) prompting to let the LLM think before it chooses an arm. 3) Instead of minimizing regret over a long time horizon, we instead work on the bandit best arm selection problem, where the LLM gets to choose arms and observe rewards for 20 turns, and then is prompted to choose what it thinks is the arm with the highest mean reward. This is done mainly to control for context length when employing COT, as we could not run inference for more than 20 turns without running into computational issues, and the observed regret between multiple models is too small if horizon length is 20. We randomize the arm rewards at every iteration. For evaluation, we use the same bandit description as Krishnamurthy

et al. (2024), for training, we use GPT-4o-mini to generate 81 diverse scenarios that are similar to it but has randomly chosen arm names and hypothetical scenarios. We also note that if the two best arms have very close mean reward (for example, 0.7 and 0.65), then it can be very difficult to identify the best arm within 20 turns. Following Krishnamurthy et al. (2024); Nie et al. (2024), we set the mean reward of the best arm to be above a certain threshold over the mean rewards of the other arms.

Finally, all the task instructions for the agents, task environments and LLM-judges were written by GPT-4o-mini, which we report next for the sake of reproducibility.

### B.1.1. NOTE ON TASK PROMPTS

We provide the task information in the first user prompt given to the agent. The system prompt for the agent on all task groups remains the same: "You are a helpful assistant.". Our initial experiments suggested that giving the task instruction in the first user prompt was more fruitful than providing it in the system prompt, though we suggest further investigation of this phenomenon.

### B.2. Note on Text-based Games

The goal of PAPRIKA is to train an LLM agent to be better at information-seeking, and to test whether these information-seeking behavior learned from a few task groups also generalizes to a new domain. To do so, we design our own task groups that require gather information to succeed. While a lot of the task groups resemble text-based games, our focus is not on them; rather text-based games are simpler information-seeking tasks that can be solved and learned reliably by language models of 8-12B parameter range, and we expect these ideas to extend to much more complicated domains given sufficiently powerful initial models.

Text-based games are an active area of research, and we would like to mention some related works here. Hausknecht et al. (2020b) utilizes interactive fiction games as a testbed for studying language based autonomous agents and their ability to handle dynamic action spaces. While our 'Murder Mystery' task group is inspired by Hausknecht et al. (2020b), particularly *Detective*, we choose to implement it separately instead of using their task environment directly, primarily due to their implementation relying on a manual parser to extract action from the LLM's generation and relying on it to take steps in the environment. The LLMs we experimented with had difficulty outputting responses in the exact format their task environment required, and we found using GPT-4o-mini to simulate the task environment to be easier while also providing more dynamic environment responses. Future work can try to directly incorporate games from Hausknecht et al. (2020b) into PAPRIKA. Similarly, text-based task groups from Côté et al. (2019); Wang et al. (2022); Jansen et al. (2024) can provide a further set of rich environments to train and test PAPRIKA-based agents on. This is a growing field with many interesting directions, we direct the readers to Jansen (2021) for a comprehensive study.

## B.3. Comparison of action and observation spaces between the task groups

*Table 3.* Summary of the initial state received by the agent, the action, and the observation spaces on all 10 task groups.

| Task Group | Initial State | Action | Intermediate Observations |
|---|---|---|---|
| Twenty Questions | User message describing the task, and the type of the secret topic (e.g., 'a concept' or 'a famous person') that the agent needs to guess within 20 questions | A yes/no question about the secret topic the agent needs to guess | GPT-4o-mini generated yes/no answer |
| Guess My City | User message describing the task | An open-ended question about the secret city the model needs to guess | GPT-4o-mini generated answer |
| Customer Service | User message describing the task, which includes a description of the problem the customer is facing | A customer service agent's troubleshooting question that aims to identify the action that will resolve the customer's problem | GPT-4o-mini simulated customer response |
| Murder Mystery | User message describing the task and the particular muder mystery scenario, including the victim, supsects and witnesses of the crime scenario | An action the detective takes in the game, aimed at identifying the perpetrator (for example, asking a suspect where they were) | GPT-4o-mini simulated game environment, e.g, the suspect answering the detective's questions |
| Wordle | User message describing the Wordle game, nature of intermediate feedback and rules the agent needs to follow | Step-by-step thinking followed by a 5 letter word guess | Feedback related to the common letters between the correct word and the guess |
| Cellular Automata | User message describing the task, and example inputs and corresponding outputs from which the agent needs to deduce the transition rule | Step-by-step thinking and then Deduction about the transition rules in a 1D cellular automaton | Inputs, true outputs, outputs generated by the transition rule the agent deduced |
| Mastermind | User message describing the rules of the task | Step-by-step thinking about past guesses and observed feedback, followed by a 4 digit secret code guess | Feedback related to the common digits between the correct code and the guess made by the agent |
| Battleship | User message describing the task, and the initial board state of the game | A particular cell that the agent wants to hit with a missile | Whether the cell contains a ship and what type of ship, followed by the current state of the board given the agent's past actions |
| Minesweeper | User message describing the task and the initial state of the grid | A particular cell that the agent wants to declare 'mine-free' | Whether the cell is mine free, followed by the current state of the grid following rules of the game |
| Bandit Best Arm Selection | User message describing the bandit game in text, including the number of turns and the possible arm names from which the agent needs to pick arm with the best reward | A particular arm in the multi-armed bandit problem that the agent chooses | Observed reward from choosing that arm, sampled from its corresponding Bernoulli distribution with fixed but unknown mean |

Table 3 shows a summary of how the task groups differ from each other.

**B.4. Details of Individual Task Groups**

B.4.1. TWENTY QUESTIONS

For twenty questions, we provide the LLM agent with general instructions about the task, and the type of hidden topic (e.g., person, location, food etc.) that it needs to guess in the first user prompt. An example is given below.

---

**Twenty Questions Agent Prompt**

You are playing a game of 20 Questions. Your goal is to guess the name of a thing or person by asking up to 20 yes-or-no questions. After each question, you will receive an answer: 'Yes' or 'No.' Use the answers provided to refine your guesses.
Here are your instructions:
- You can ask only yes-or-no questions.
- After receiving each answer, you should adapt your questions based on the new information.
- Your goal is to guess the topic in as few questions as possible.
- If you're confident, you can make a guess before reaching 20 questions.
The game starts now. You are trying to guess a clothing. Ask your first question!

---

We use another LLM (usually GPT-4o-mini unless explicitly mentioned otherwise) to simulate the task environment that provides yes/no answers. This LLM receives the secret topic, and generates answers to the agent's questions in relation to this topic. An example system prompt for the task environment is given below.

---

**Twenty Questions Environment Prompt**

You are the environment for a game of 20 Questions. You will be given a topic (e.g., cat, deer, Abraham Lincoln), and your role is to answer 'Yes' or 'No' to questions about the topic. Respond strictly with 'Yes' or 'No' only, based on the truth of the matter. Do not provide any additional information. If the user guesses the correct answer, respond with 'Goal reached.' Enforce a strict match before saying 'Goal reached.', for example, if the user guesses 'Is the occupation Engineer?' and the topic was software engineering, say 'No'. However, if the user guesses 'Is the occupation a type of engineering?', say 'Yes'. Wait for the user's questions and respond accordingly.
For example:
- User: 'Is this person alive?'
- You: 'No'
- User: 'Is this a living creature?'
- You: 'Yes'
The player is trying to guess a clothing, and the correct answer is Gloves.

---

To prevent task environment hacking, we additionally perform string matching to ensure the response from the task environment is yes, no or 'Goal reached'. We discard trajectories where the task environment responds with something else everytime within 5 attempts. Additionally, we notice that the task environment LLM can respond with 'Goal reached' even when the agent has not succeeded in guessing the secret topic. We use an additional API call to GPT-4o-mini as a judge to filter these trajectories. An example system prompt for this LLM judge is as follows:

Twenty Questions Judge Prompt

You are an expert judge for the game of 20 questions. You will be given a question asked by the player, and you need to judge if it successfully guesses the correct topic.

In particular: Check that the player has precisely guessed Gloves in their question. Reply with <VALID> or <NOTVALID>. Reply with <VALID> if and only if the given question successfully ends the game by guessing the correct clothing, which is: Gloves, otherwise reply with <NOTVALID>.

Example 1: Topic to Guess: Car
Player: Is the invention a car?
Answer: <VALID>
Example 2: Topic to Guess: Car
Player: Does the invention have wheels?
Answer: <NOTVALID>
Example 3: Topic to Guess: Apple, Inc.
Player: Does this company produce IPhones?
Answer: <NOTVALID>
The conversation begins here:
Agent: Based on the fact that the clothing is worn for warmth and on the hands, I'm going to guess that the clothing is a glove.
(End of Agent Turn)
Now judge whether the player has successfully guessed the correct clothing, which is Gloves in this particular game. Reply with <VALID> only if the player has guessed Gloves in the question, otherwise reply with <NOTVALID>. Note that guessing a particular characteristics of Gloves is not enough, the player needs to arrive at the final answer in order for you to reply with <VALID>.
Answer:

B.4.2. GUESS MY CITY

An example prompt for the agent is listed below:

---

**Guess My City Agent Prompt**

You are playing a game called 'Guess My City.' Your task is to guess the home city of the other player within 20 questions.
You must follow these rules:
1. You may ask open-ended questions about the city's characteristics, such as geography, culture, landmarks, cuisine, climate, or other unique traits.
2. Do not ask directly for the name of the city or country.
3. Use the answers from the other player to strategically decide the next question, your goal is to arrive at the correct city with as few questions as possible.
4. After gathering enough information, you can attempt to guess the city, but each guess counts as one of your 20 questions.
Example questions: 'What is your favorite cuisine from your home city?', 'Is your home city located near the coastline?', 'What kind of landmarks is your city known for?' Your goal is to identify the city through deduction, pattern recognition, and strategic questioning.
4. When you are confident, make a guess, by mentioning the name of the city and the country it is in, for example, 'Is the city Jakarta, Indonesia?'
The game starts now, and start asking your questions.

---

We use GPT-4o-mini to simulate the task environment. We provide the name of the city the agent needs to guess to the environment and instruct it to generate answers related to this target city, without giving away the name of the city unless the agent guesses it. An example system prompt for the task environment is listed below:

---

**Guess My City Environment Prompt**

You are the environment in a game called 'Guess My City.' You will be given a specific home city (e.g., London, United Kingdom) and you have to respond to the player's questions to help them guess this city.
Follow these rules:
1. Answer the agent player's questions honestly and correctly (i.e., the answers need to be true given the home city given to you at the start of the game), but do not mention the name of your city or its country explicitly. However, you can verify the player's guess about a particular city/country. For example, if the player asks, 'What is your home country?' You should not respond to this question. However, if they ask 'Is your home country United Kingdom?' You should reply with the correct answer based on the home city given to you.
2. Your answers should provide useful clues about your city's geography, culture, landmarks, cuisine, climate, or other unique characteristics.
3. Be as informative as possible to guide the agent player within the rules. Example answers: If the agent asks about cuisine, you could say: 'My city is famous for its seafood dishes and has a vibrant food market scene.' If the agent asks about geography, you might say: 'My city is located near a major river and has a historic port.'
4. You should respond with 'Goal reached' if the player has successfully guessed your home city. Ignore minor variations in the spelling of the city, but only if these variations are common.
Remember, you are here to help the agent guess your city through clues, but you must not reveal the city's name or its country directly UNDER ANY CIRCUMSTANCES. If the player asks such a question, reply that you cannot give away the name, and can only confirm the name if the player guesses it.
The game starts now, and the player is trying to guess the city Surat, India

---

To prevent the task environment from getting hacked, we use GPT-4o-mini as a judge similar to twenty questions. An example system prompt for the judge is listed below.

Guess My City Judge Prompt

You are the judge in a game called 'Guess My City'. Your role is to verify if the player has won the game by correctly guessing the city chosen by the environment (env) player.

To determine this, follow these rules:

1. Check the agent player's question to ensure they have not directly asked the environment for the name of the city or country. Any question explicitly requesting the city's name, such as 'What is the name of your city?' or 'What country is your city in?' violates the rules.

2. Verify that the player has explicitly mentioned the correct name of the city in their guess. Ignore minor variations in the spelling of the city (for example, Kiev and Kyiv refer to the same city). If these two conditions are met, the player wins the game. Otherwise, the agent has not won.

3. Your role is purely evaluative, ensuring adherence to the rules and confirming or denying the win based on the criteria above.

4. Reply <VALID> if the player has successfully guessed the city, and followed the specified rules. Otherwise, reply <NOTVALID>

The conversation begins here:

Agent: I'm not giving up yet. Based on your previous answers, I'm going to try again. Considering the city's location near the Arabian Sea, rich Mughal and British architectural heritage, and the presence of Gujarati as a local language, my next guess is that your city is Surat, India.

(End of Agent Turn)

Now judge whether the player has successfully guessed the correct city, which is Surat, India in this particular game. Reply with <VALID> only if the player has guessed Surat, India (or other variations in name, if both names refer to the same city) in their response, otherwise reply with <NOTVALID>. Ignore variations in the name of the city if the city is known by both names (for example, Kyiv and Kiev). Note that guessing a particular characteristics of this city is not enough, the player needs to successfully guess the correct city by name in their response in order for you to reply with <VALID>.

Answer:

B.4.3. CUSTOMER SERVICE

For this task group, we require the agent to act as a customer service agent, with the following prompt:

---

**Customer Service Agent Prompt**

You are going to role-play as a customer service agent and you have to help a customer resolve their issue. Your goal is to gather enough information to diagnose the problem and provide solution.
Your instructions are the following:
1.You will need to ask targeted questions or suggest particular actions to the customer to gather the necessary details.
2. The customer may not be technically inclined, so keep your language simple and clear.
3.Avoid making assumptions — ask specific questions to determine the potential causes. You should guide the customer through basic troubleshooting steps and gather data on the situation.
4. Refine your questions in a strategic way based on the customer's responses for earlier questions.
5.You should ask questions in an efficient manner, to make the customer satisfied and resolve their problem as quickly as possible. You should also keep your responses short and concise.
6. If the customer mentions a specific product they are using (for example, ABC electronics), then you are the customer support agent for that product/company, i.e., you represent that product or company and have to take appropriate actions without referring the customer to somewhere else.
Your specific scenario is this: The customer's refrigerator is beeping continuously and is not cooling effectively. Please start helping the customer now by asking your first question.

---

Similar to the two prior task groups, we use another LLM (usually GPT-4o-mini) to simulate the task environment. We specifically instruct it to act as a customer without much technical knowledge.

---

**Customer Service Environment Prompt**

You are going to role-play as a customer experiencing a specific issue.
You must adhere to the following guidelines:
1. Only respond to the agent's questions and provide relevant information when prompted.
2.Your responses should be concise and reflect a typical, non-technical customer's knowledge level.
3. You are going to know what the solution for the specific issue is, but do not give away this information at any point. Your goal is to observe if the customer service agent can ask relevant questions and arrive at the specific issue.
4.When the agent asks you about a potential solution, you should determine what the outcome would be based on your knowledge about the true underlying problem, and let the agent know the result. For example, if your car is not starting because of lack of fuel, and the agent suggests you to check the fuel level in your car, it should fix your issue. Use your reasoning ability to decide whether the suggested solution works, given that the solution to your problem is: The refrigerator door is slightly ajar, and properly closing it will stop the beeping and restore cooling.
5. Reply with 'Goal reached' (and nothing more) if the agent has reached or solved your particular issue.
Your role-play starts now. The particular problem you, as the customer, are facing is: The customer's refrigerator is beeping continuously and is not cooling effectively., and the solution to this problem is: The refrigerator door is slightly ajar, and properly closing it will stop the beeping and restore cooling.. Note that you should never give away or tell the agent what will solve your particular problem, your goal is to see if the agent can come up with the potential solution themselves.

---

Finally, we use an LLM judge at every turn similar to twenty questions and guess my city, to filter trajectories that may have hacked the task environment.

## Customer Service Judge Prompt

You are the judge of a customer service agent trying to help a customer in a particular scenario. You would be given the the agent's response, and you have to verify whether the agent has been able to solve the problem of the customer correctly.

The particular scenario the customer is facing is: The customer's refrigerator is beeping continuously and is not cooling effectively., and the solution to their problem is: The refrigerator door is slightly ajar, and properly closing it will stop the beeping and restore cooling.

Reply with <VALID> if the agent is successful in suggesting the correct solution, otherwise reply with <NOTVALID>.

The conversation begins here:

Agent: A steady beep can indicate a specific issue.

Have you checked the temperature settings on your refrigerator? Is it set to the correct temperature, and are the doors closed properly?

(End of Agent Turn)

Now judge whether the agent has been successful in making the correct suggestion to solve the customer's problem. Use your reasoning ability to decide whether the agent's response would solve the customer's problem, which is The refrigerator door is slightly ajar, and properly closing it will stop the beeping and restore cooling.

For example, if the customer's car is not starting because of a lack of fuel, and the agent suggests to check the fuel level in the car, it should fix the issue. Reply with <VALID> if they have been successful, otherwise reply with <NOTVALID>.

Answer:

B.4.4. MURDER MYSTERY

For this task group, we prompt the LLM agent with a particular murder mystery scenario. An example prompt is given below.

---

**Murder Mystery Agent Prompt**

You are playing the role of a detective in a murder mystery game.

The setup for the game is:

1.You will be provided with a scenario describing a crime and its key elements. Your goal is to solve the mystery by asking questions, examining evidence, and drawing logical conclusions.

2. For every action you take or question you ask, you will receive feedback from the game.

3. Your questions and actions should be precise and logical, aimed at uncovering clues, verifying alibis, and piecing together the sequence of events. You should stretegically choose the next action, given the information you have already obtained from the game, and choose actions that lets you catch the culprit as quickly as possible.

4. You can only take a single action at every turn.

5. You have to consider all pieces of information, and scrutinize all the characters in the game, including the witnesses or background characters, since the true culprit maybe a witness or a background character, and might not always be one of the primary suspects declared at the beginning of the game. Do not focus on any character too early in the game, rather try to see if anyone's statements are contradictory.

6. You should always gather enough information before making a decision — try not to make a mistake! You should also keep your mind open about who can be the true culprit and try to be information seeking, without being too narrowed down on one suspect too quickly.

7. Once you believe you have enough evidence, you may state your conclusion about the case, which will terminate the game.

The game starts now. The particular scenario you have is: You are a detective investigating the death of Aiko Nakamura, a curator found dead during the exhibition of a centuries-old scroll at Tokyo's National Museum of Art. The suspects are her protégé Hiroshi, sponsor Ms. Tanaka, and rival curator Dr. Saito. Witnesses include Akiko, a janitor who was cleaning nearby. Investigate the scroll's history and the black market connections to find the murderer.

---

The corresponding environment prompt for the same task is as follows (given to GPT-4o-mini to simulate the task environment):

---

**Murder Mystery Environment Prompt**

You are an advanced AI responsible for simulating a challenging and immersive murder mystery game environment. You are provided with the full details of the scenario, including the characters, their backstories, motives, alibis, the sequence of events, and the true culprit. Your task is to respond to the actions and inquiries of the detective player in a manner that maintains the game's logic and adds complexity to the mystery.

Your instructions are:

1. Provide only the information directly prompted by the detective's actions or questions, avoiding any unnecessary hints or details. Keep your responses ambiguous and concise to encourage deeper investigation.

2. If the detective inspects an item, describe its condition, appearance, and only the most relevant clues, avoiding overly revealing details unless the detective's inspection is exceptionally thorough or precise. If they interview a character, simulate realistic behavior and speech based on the character's personality, motivations, and knowledge. Characters will actively avoid implicating themselves and may misdirect suspicion.

3. Allow for logical consequences of the detective's actions, such as making certain items available for inspection, unlocking additional questions, or creating natural developments in the storyline. Maintain consistency in responses and adjust dynamically to the detective's progress.

4. When the detective declares their final accusation, reply with 'Goal reached', and do not say anything else.

5. Characters will never explicitly admit to guilt but will respond realistically under pressure. Create a dynamic and challenging experience by balancing character responses between truth, omission, and plausible misdirection.

6. Always strive to make the game enjoyable and intriguing while revealing the absolute minimum information required for the detective to progress logically.

The game begins now. The specific scenario for this session is: In Tokyo's National Museum of Art, a renowned curator, Aiko Nakamura, is found dead during the exhibition of a centuries-old scroll. The suspects are her protégé Hiroshi, who argued over her criticism of his work; a corporate sponsor, Ms. Tanaka, who pushed for the scroll's relocation; and a rival curator, Dr. Saito, who claimed the scroll was a forgery. The true culprit is Akiko, a janitor, who killed Aiko to prevent her from exposing a secret black market deal involving the scroll. Key evidence includes Akiko's fingerprints on the exhibit case, forged documents in her locker, and surveillance footage showing her entering the gallery after hours.

---

Finally, similar to the prior task groups, we also use GPT-4o-mini as a judge to verify the task success rewards, with the following example prompt.

---

**Murder Mystery Judge Prompt**

You are the judge in a murder mystery game. Your role is to evaluate the detective's final action and determine if their conclusion is correct.

You will be provided with two inputs:

1.the true crime scenario, which includes the actual culprit. The true game scenario in this round is this: In Tokyo's National Museum of Art, a renowned curator, Aiko Nakamura, is found dead during the exhibition of a centuries-old scroll. The suspects are her protégé Hiroshi, who argued over her criticism of his work; a corporate sponsor, Ms. Tanaka, who pushed for the scroll's relocation; and a rival curator, Dr. Saito, who claimed the scroll was a forgery. The true culprit is Akiko, a janitor, who killed Aiko to prevent her from exposing a secret black market deal involving the scroll. Key evidence includes Akiko's fingerprints on the exhibit case, forged documents in her locker, and surveillance footage showing her entering the gallery after hours.

2. The detective's last response.

Compare the detective's action to the true crime scenario. Reply with <VALID> or <NOTVALID> strictly. You should reply <VALID> if and only if the detective has successfully caught the true culprit in their response, and otherwise respond with <NOTVALID>.

The conversation begins here:

Agent: Akiko's panic and desperation suggest that she's hiding something, and her story about hearing Aiko talking to someone in the back room raises more questions than answers. I believe I have enough evidence to make an accusation. Based on the inconsistencies in Akiko's story, her nervous behavior, and her knowledge of the hidden compartment in the scroll, I believe Akiko is the culprit. I accuse Akiko of murdering Aiko Nakamura.

(End of Agent Turn)

Now judge whether the detective has successfully accused the person who truly committed the crime in their last response. Here is the true crime scenario: In Tokyo's National Museum of Art, a renowned curator, Aiko Nakamura, is found dead during the exhibition of a centuries-old scroll. The suspects are her protégé Hiroshi, who argued over her criticism of his work; a corporate sponsor, Ms. Tanaka, who pushed for the scroll's relocation; and a rival curator, Dr. Saito, who claimed the scroll was a forgery. The true culprit is Akiko, a janitor, who killed Aiko to prevent her from exposing a secret black market deal involving the scroll. Key evidence includes Akiko's fingerprints on the exhibit case, forged documents in her locker, and surveillance footage showing her entering the gallery after hours. Respond with <VALID> if the detective is successful, otherwise reply with <NOTVALID>. Note that the detective has to announce their accusation in order for you to respond with <VALID>, and merely confronting a character is not enough.

---

B.4.5. WORDLE

For wordle, we use a hardcoded program as the task environment, that generates intermediate observations and eventual task reward. The LLM agent playing wordle receives the instructions for this task in its prompt. Furthermore, we prompt it to use chain-of-thought before generating a final response:

---

**Wordle Agent Prompt**

You are playing a game of Wordle. Your goal is to guess the secret five-letter word within six attempts. After each guess, you will receive feedback in the form of a series of statements describing how the letters in your guess compare to the secret word. Each statement corresponds to a letter in your guess:
- 'First letter is correct and in the correct position in the target word' means the letter is correct and in the right position.
- 'First letter exists in the target word, but in a different position' means the letter is correct but in the wrong position.
- 'First letter does not exist in the target word' means the letter is not in the word at all.
Use this feedback to refine your guesses and try to guess the secret word within six attempts. You should try to strategically choose your guesses based on prior guesses (if any) and corresponding feedback you received, so that you can guess the secret word as quickly as possible.
You have to refine your guess based on this provided feedback. Keep guessing until you either guess the word correctly or use up all your attempts.
Please try to be concise. Format your response in the following way: `<Think>` Any step-by-step, short and concise thinking to strategically determine the next guess for the secret word `</Think>`
`<Answer>` your guess of what the word should be `</Answer>`
The game begins now, please make your first guess about the secret five-letter word!

---

We also provide an example of the task environment feedback: given the secret word 'toast' and the agent's guess 'boost', we generate the following feedback:

---

**Wordle Task Environment Feedback**

First letter, b, is not in the target word
Second letter, o, is correct and in the correct position in the target word
Third letter, o, exists in the target word but in a different position
Fourth letter, s, is correct and in the correct position in the target word
Fifth letter, t, is correct and in the correct position in the target word
Make your next guess about the hidden word. Please try to be concise. Format your response in the following way: `<Think>` Any step-by-step, short and concise thinking to strategically determine the next guess for the secret word `</Think>`
`<Answer>` your guess of what the word should be `</Answer>`

---

### B.4.6. Cellular Automata

For this task group, we want an LLM to be able to infer the transition rule of 1D elementary cellular automation by observing the inputs and outputs of its previously inferred transition rule, plus the correct outputs for the same inputs if the inferred transition rule was wrong. Recall that for 1D cellular automation, we have binary strings consisting of '1' and '0' as a state, e.g., '111010' can be a state. Each '1' and '0' are referred to as a cell within the state. We also have a transition rule that defines how each cell would transform in the next state given its left and right neighbor. For any cell $c$, we call (left neighbor, cell, right neighbor) the neighborhood of $c$.

For example, consider the following transition rule:

| Neighborhood of center cell | 111 | 110 | 101 | 100 | 011 | 010 | 001 | 000 |
| --- | --- | --- | --- | --- | --- | --- | --- | --- |
| New state for center cell | 0 | 1 | 1 | 0 | 1 | 1 | 1 | 0 |

Here $111 \rightarrow 0$ implies that if a cell is '1' and both its left and right neighbors are '1', then the cell will become '0' in the next time step. We adopt the convention that for the left-most cell in the state, we consider the right-most cell as its left neighbor, and similarly for the right-most cell, we consider the left-most cell as its right neighbor.

Now we would show an example for how to calculate the output state given the input state and the transition rule. Assume the input state is '10110', and we want to apply the transition rule from above. Then we compute the next state as follows:

1. The first cell is 1, the last cell is 0 (which will be considered as the first cell's left neighbor), and the second cell is 0. So the neighborhood of the first cell is '010'. For this neighborhood, we have the transition rule $010 \rightarrow 1$, so the first cell remains 1

2. Similarly, the neighborhood of the second cell is 101. Now $101 \rightarrow 1$, so the second cell becomes 1 from 0

3. $011 \rightarrow 1$, so the third cell remains 1

4. $110 \rightarrow 1$, so the fourth cell remains 1

5. $101 \rightarrow 1$, so the fifth cell becomes 1 from 0

Therefore, the next state becomes '11111' from '10110'.

Note that there are 256 possible transition rules. In the first user prompt, we choose a few random binary strings as input states. We also pick one of the 256 transition rules randomly and use it to generate the next states given the input states and this transition rule. We then provide the LLM with these (input state, output state) pairs, and ask it to infer the transition rule. There can be multiple correct transition rules that generate the same output states from the input states (since the input states may not have all 8 possible neighborhood configurations), so we declare task success if the guessed transition rule by the agent generates outputs that match the given output states (we do not require the guessed transition rule to exactly match the hidden transition rule, as long as it generates correct outputs from the given inputs). If the LLM generated transition rule does not generate the correct output for all given inputs, we provide it with the outputs its predicted rule would generate and ask it to try again. This is intended to simulate the ability to code a function given inputs and desired outputs from the user, and then refine previously written code using feedback from an available interpreter.

An example instruction prompt for this task group is given next.

---

Cellular Automata Agent Prompt

You are a reasoning assistant participating in a game where you must deduce the hidden rule governing a 1D cellular automaton. In each round, you are provided with 3 inputs (the initial state of the automaton) and the corresponding outputs after applying the hidden rule for one step. Your task is to analyze the input-output pairs and deduce the hidden rule that governs the automaton's behavior. If your guessed rule generates the correct outputs for the given inputs, you win the game. If your guess is incorrect, the game will provide you with the outputs generated by your guessed rule for the same inputs as feedback. Use this feedback to refine your guess in subsequent rounds. Your goal is to try to guess the correct hidden rule as quickly as possible.

The rule governs the behavior of each cell in the automaton based on its state and the state of its immediate neighbors (left, center, and right). There are 8 possible configurations of these states, each represented as a 3-bit binary number (e.g., '111', '110', '101', etc.). For the first and last cells, we warp around the edges, i.e., the left neighbor of the first cell is last cell, and the right neighbor of the last cell is the first cell. Your guess must specify the next state for each configuration in the following format:

'<Think> step-by-step thinking to deduce the correct hidden rule </Think>
<Answer>
<rule> 111: next state </rule> <rule> 110: next state </rule> <rule> 101: next state </rule>
<rule> 100: next state </rule> <rule> 011: next state </rule> <rule> 010: next state </rule>
<rule> 001: next state </rule> <rule> 000: next state </rule> </Answer>'

### Explanation of the format:
- '<rule> 111: 0 </rule>' means if the current cell and both of its neighbors (left and right) are in state 1, then the current cell will transition to state 0 in the next iteration.
- Similarly, '<rule> 110: 1 </rule>' means if the left and center cells are in state 1 and the right cell is in state 0, then the current cell will transition to state 1 in the next iteration.

### Example Round:
**Input and Output Provided:**
Input 1: 0 1 1 1 1 0
Output 1: 1 1 0 0 1 0
**Your Response:**
'<Think> Based on the provided example, I observe that cells transition to state 0 when surrounded by 1s, and cells surrounded by exactly two active neighbors transition to state 1. Using this reasoning, I deduce the following rule: </Think>
<Answer>
<rule> 111: 0 </rule> <rule> 110: 1 </rule> <rule> 101: 1 </rule> <rule> 100: 0 </rule>
<rule> 011: 1 </rule> <rule> 010: 0 </rule> <rule> 001: 1 </rule> <rule> 000: 0 </rule>
</Answer>
If your guessed rule does not produce the correct outputs, you will receive feedback. For instance:
Input: 0 1 1 1 1 0
Your Output: 1 0 0 1 0 0
Use this feedback to refine your rule in the next round. Continue iterating until your guessed rule generates outputs matching the true outputs for the provided inputs. Aim to win the game by accurately deducing the hidden rule as quickly as possible.
The game begins now, and your (input, output) pairs are:
Input 1: 0 0 0
Output 1: 0 0 0
Input 2: 1 1 1 1 1 0 0 0 1
Output 2: 0 0 0 0 0 1 0 1 1
Input 3: 1 0 0 1 1 1 0 0
Output 3: 1 1 1 1 0 0 1 1

When the agent makes a wrong guess, it receives feedback from the task environment as follows:

---

**Cellular Automata Environment Feedback**

Sorry, the automation rule you guessed does not generate the correct outputs for all the given inputs. I will give you the outputs from the rules that you gave last time. Please use them to refine your guess about the automation rule.
Input 1: 0 0 0
True Output 1: 0 0 0
Output generated by the last rule you gave: 0 0 0
Input 2: 1 1 1 1 1 0 0 0 1
True Output 2: 0 0 0 0 0 1 0 1 1
Output generated by the last rule you gave: 0 0 0 0 0 0 0 0 0
Input 3: 1 0 0 1 1 1 0 0
True Output 3: 1 1 1 1 0 0 1 1
Output generated by the last rule you gave: 0 0 0 0 0 0 0 0
Make your next guess about the hidden rule. Format your response in the following way:
'<Think> step-by-step thinking to deduce the correct hidden rule </Think>
<Answer>
<rule> 111: next state </rule> <rule> 110: next state </rule> <rule> 101: next state </rule>
<rule> 100: next state </rule> <rule> 011: next state </rule> <rule> 010: next state </rule>
<rule> 001: next state </rule> <rule> 000: next state </rule> </Answer>'
Keep your thinking concise.

---

B.4.7. MASTERMIND

For mastermind, we have a secret 4-digit code (each digit can be anything between 0 and 9), and ask an LLM agent to guess it. The agent starts with a 4-digit guess, and the task environment provides feedback in terms of:

- **Exact matches**: How many of the digits in the guess are also in the target secret code, and exactly in the same position? In other words, the number of exact matches reflects the number of positions that are exactly the same between the guess and target code.

- **Partial matches**: Discounting the exact match digits, how many of the other digits in the guess code are in the target secret code? In other words, the number of partial matches reflect the digits in the guessed code that are in the secret code but in different positions.

For a concrete example, assume the secret code is '1706', and the LLM at a particular iteration has guessed '1608'. Then it would receive the following feedback:

- There are two exact matches. The two exact matches are 1 and 0, in first and third position, though this information would not be revealed to the LLM, it must reason about this by looking at the information from all previous turns.

- There are one partial match. This is the digit 6, which is in the target secret code, but in a different position. The LLM would only receive the information that there is 1 partial match, and not the information about which digit corresponds to that match.

Now that we have explained the rules of the task, we would provide the instruction prompt describing the task to the LLM agent, which also describes the complete rules for this task:

---

**Mastermind Agent Prompt**

You are an AI playing the game Mastermind with digits. The objective of the game is for you, the codebreaker, to guess a secret code of 4 digits, where each digit ranges from 0 to 9. The code is created by the codemaster and can include repeated digits.

The gameplay proceeds as follows:

1. You make a guess by proposing a 4 digit code. You should state your guess as 4 digits separated by a space.

2. After each guess, the codemaster provides feedback in the form of two numbers:

- 'Exact matches' – The number of digits in your guess that are correct and in the correct position.

- 'Partial matches' – The number of digits (distinct from exact matches) in your guess that are correct but in the wrong position.

Given this feedback, DO NOT simply assume any particular digit is an exact or partial match or not in the secret code, you should have strong reasoning based on obtained feedbacks to make deductions on particular digits.

3. Using this feedback, you refine your future guesses, aiming to deduce the secret code.

Rules for feedback:

- Each digit in the secret code can only contribute to feedback once.

- If a digit is correct but occurs more times in your guess than in the code, the extra occurrences are ignored for partial matches.

The game ends when you correctly guess the code, achieving 4 exact matches.

Your goal is to refine your guess about the secret code using the feedback provided by the codemaster, and strategically choose your next guess so as to be able to guess the correct code as quickly as possible.

The game starts now, make your first guess! You should format your response as: `<Think>` Any step-by-step, short and concise thinking to determine what the next guess should be `</Think>`

`<Answer>` your guess on the 4 digit code `</Answer>`

---

Below is an example of hardcoded task environment feedback, when the true secret code is '5959', and then LLM agent has guessed '5789':

---

**Mastermind Task Environment Feedback**

Your last guess has 2 exact matches with the secret code. In other words, exactly 2 digit(s) in your last guess, 5 7 8 9, are in the correct position in the secret code. (We won't reveal the particular digits within your guess that are exact matches, they can be any digit within your guess) Your last guess also has 0 partial matches. In other words, 0 digits in your guess, 5 7 8 9, are in the secret code, but in the wrong position. (We won't reveal which digits within your guess are partial matches, they can be any, you must deduce them with reasoning and further guesses and feedbacks.) Now make your next guess about the secret code. Please format your response as: `<Think>` Any step-by-step, short and concise thinking to determine what the next guess should be `</Think>`
`<Answer>` your guess on the 4 digit code `</Answer>`

---

B.4.8. BATTLESHIP

We employ a modified version of the battleship game here as one of our task groups: https://en.wikipedia.org/wiki/Battleship_(game). The main modifications are:

- We make an entirely text-based version of this game for the purpose of our paper.

- We want to test strategic exploration and decision-making capabilities of LLMs without having to worry about an adversary, so we make the game single player, where the agent just needs to find and sink all of the enemy ships in the grid within a certain number of turns to achieve victory (and does not need to consider their own ships getting sunk by an adversary). We leave the two-player version of this game for future work.

In our version of the game, we start with a $N_1 \times N_2$ grid, where we place 3 ships: a carrier requiring 5 contiguous horizontal or vertical cells within the grid, a battleship requiring 4 cells, and a destroyer requiring 2 cells. The ships are placed randomly at every iteration, and the ships locations are hidden from the agent. Imagine the true board state looks like following:

|   | 1 | 2 | 3 | 4 | 5 |
|---|---|---|---|---|---|
| A | Carrier | Carrier | Carrier | Carrier | Carrier |
| B | Battleship | | | | |
| C | Battleship | | | | |
| D | Battleship | | | Destroyer | Destroyer |
| E | Battleship | | | | |

The co-ordinates in the grid are marked by row identifiers (letters starting from 'A') and column identifiers (numbers starting from 1). For example, in the above board, the carrier is placed on cells A1, A2 upto A5. At every turn, the agent gets to choose a particular cell (for example, 'C2') to hit with a missile. It then receives the following feedback from the task environment:

- If the cell was targeted in an earlier turn, nothing happens, and the agent is informed about this.

- If the cell was not targeted before and is empty, then the agent is informed that their choice was a miss.

- If the cell was not targeted before and has a ship in it, then the task environment informs the agent that their choice of the cell resulted in a hit. It also announces what type of ship was hit by the agent. If the agent has hit all the cells in the grid pertaining to a particular ship, then the task environment also announces that the particular ship has been sunk.

- If the agent has sunk all 3 ships, then the task results in success. Otherwise, if the all of the allowed number of turns has passed and there is at least one ship remaining in the grid, then the task ends in failure.

After every turn, the agent gets an updated view of the board with the hits and misses clearly marked out. For example, if we mark misses with an 'M', successful hits with an 'X', and hidden cells with an '.', and if the agent chooses to target C2 and A1 in the first two turns respectively, then the corresponding board that the agent will observe at the beginning of the third turn looks like the following:

|   | 1 | 2 | 3 | 4 | 5 |
|---|---|---|---|---|---|
| A | X | . | . | . | . |
| B | . | . | . | . | . |
| C | . | M | . | . | . |
| D | . | . | . | . | . |
| E | . | . | . | . | . |

**In order to be successful at battleship, agents need to balance between exploration and exploitation similar to the bandit setting, but without well-known optimal algorithms.** At the start of the game, an agent needs to explore the board effectively to find ship locations, and once it has a hit a particular ship, it would need to exploit around that particular cell to find all cells pertaining to the ship to be able to sink it completely.

Next, we provide the description of the task given to the LLM agent at the start of the task, explaining the rules:

---

**Battleship Agent Prompt**

You are playing a single-player version of the Battleship game. Your objective is to sink all ships on the board in as few attempts as possible, with a maximum of 20 attempts. The game is played on a grid size: 6 x 6 grid, and the board uses the following symbols:
- '.' represents a hidden cell that has not been hit.
- 'X' represents a cell where you successfully hit a ship.
- 'M' represents a cell you have hit previously, which was a miss, i.e., there were no ships in that cell.
Rules:
1. There are 3 ships hidden on the board:
- Carrier: size 5
- Battleship: size 4
- Destroyer: size 2
2. Ships are placed either horizontally or vertically and do not overlap.
3. On each turn, choose a cell to attack by providing its coordinates (e.g., A1, B3).
4. If you hit a ship, the cell will change to 'X'.
5. If you miss, the cell will change to 'M'.
6. The game ends when all ships are sunk or after 20 attempts. After every attempt, I will show you the current board state.
Use logic to deduce the possible locations of remaining ships as the board fills in.
Remember to focus on sinking the ships efficiently while minimizing wasted turns.
Cells are represented with the row being denoted with a letter, starting from A and and so on, and the columns being denoted by 1, 2, 3, and so on.
The cell in the first row and column is labeled A1, the second cell in the second column is labeled B2, etc.
You should format your response as the following:
<Think> Any step-by-step, short and concise thinking to strategically determine which cell you should hit next </Think>
<Answer> the cell you chose to hit </Answer>
The game begins now, with the board looking like the following:
1 2 3 4 5 6
A . . . . . .
B . . . . . .
C . . . . . .
D . . . . . .
E . . . . . .
F . . . . . .
Please make your first move!

---

In the above example, there was no ships placed at D1, and if the agent chooses to target it, it will give the following task environment feedback:

---

**Battleship Environment Feedback Example 1**

Miss at D3. There is no ship in this co-ordinate. Here is how the board looks now:
1 2 3 4 5 6
A . . . . . .
B . . . . . .
C . . . . . .
D . . M . . .
E . . . . . .
F . . . . . .
Please make your next move.

---

After a few turns, the agent chooses to target the cell A2, which has a carrier secretly placed in it. Then it receives the following feedback:

---

Battleship Environment Feedback Example 2

Hit at A2! You have hit a Carrier, which occupies 5 cells in the grid.
Here is how the board looks now:
1 2 3 4 5 6
A M X . . . .
B . . . . . .
C . . . . . .
D . M M . . .
E . . . . . .
F M . . . . .
Please make your next move.

---

The other types of feedback are provided in a similar fashion, which we omit here for the sake of brevity.

B.4.9. MINESWEEPER

We adopt a text-based version of minesweeper ([https://en.wikipedia.org/wiki/Minesweeper_(video_game)](https://en.wikipedia.org/wiki/Minesweeper_(video_game))), a logic puzzle game, as a task group for PAPRIKA. The task rules are as follows:

**1. Setup**

The game board is an $m \times n$ grid. Each cell is either empty or contains a mine. Mines are placed randomly and remain hidden until revealed. Hidden cells are represented with '#'. Number of mines is also chosen randomly.

**2. Cell Reveal**

The agent selects a cell to reveal. If the cell contains a mine, the game ends. The first cell the agent chooses to reveal has no mines, and mines are only placed randomly along the grid after the first cell has been chosen by the agent to be revealed, excluding the first chosen cell. If the cell is empty, it displays a number indicating the count of mines in its 8 adjacent cells (or '*' if the number is 0).

**3. Numbered Cells**

A revealed cell shows a number between 1 and 8, and '*' if it has no mines and none of its neighbors also has mines. The number represents how many mines are adjacent to that cell (including diagonals).

**4. Reveal Mechanism**

If a revealed cell has a zero, it automatically reveals all adjacent cells. This process continues recursively for adjacent '*' cells. The chain stops when cells with non-zero numbers are reached.

We will give an example game-play here to make the rules clearer. Imagine we start with a $5 \times 5$ grid. The initial board will look like the following:

| # | # | # | # | # |
|---|---|---|---|---|
| # | # | # | # | # |
| # | # | # | # | # |
| # | # | # | # | # |
| # | # | # | # | # |

Next, the agent chooses to reveal the cell at row 2, column 2 (0-indexed). The task environment then randomly places mines, and produces the following board after executing the reveal mechanism above:

| # | # | 1 | * | * |
|---|---|---|---|---|
| 1 | 1 | 1 | * | * |
| * | * | * | * | * |
| * | 1 | 1 | 1 | * |
| * | 1 | # | 1 | * |

It is easy to see that the cell at (4, 2) and (0, 1) have mines. So the only cell left without a mine is (0, 0), and if the agent chooses to reveal it, then the task ends with success. If the agent chooses to reveal (4, 2) or (0, 1), then the task ends with failure. If the agent chooses to reveal any other cell, nothing happens and just a turn gets wasted.

Now we provide an example instruction prompt given to the agent for this task group, describing the rules of this task:

---

**Minesweeper Agent Prompt**

You are a playing the game of Minesweeper. You will be given a two dimensional board that looks like:
# # #
# # #
# # #
with each row of the board presented sequentially, and different rows separated by newline. The game board is represented by a grid of characters:
(a) '#' indicates a hidden cell; in other words, you do not whether this cell has a mine in it or not,
(b) '*' indicates a revealed empty cell, i.e., a cell marked with '*' has been revealed and it does not have any mines, and
(c) digits (1 through 8) indicate the count of mines in adjacent cells (for example, if a cell has digit 3 on it, it means 3 out of 8 of its adjacent cells have mines, but it does not tell you if this particular cell has mines or not).
You will be given the current board state from a user. Your task is to analyze it, apply standard Minesweeper logic, and suggest the next move(s).
The rows and columns in this game use 0-based indexing, i.e., the first row is indexed by 0, the second row is indexed by 1, and so on.
Provide step-by-step, short and concise reasoning for how you identify any guaranteed safe cells and guaranteed mines, then propose the final move.
If multiple moves are possible, choose the most logical option.
Follow these instructions carefully and maintain consistency with the rules of Minesweeper. Your goal is to reveal all the empty cells, without revealing any of the cells that has mines. You should make logical deductions to avoid cells you think can have mines, while choosing the next cell to reveal.
You should format your response as follows:
<Think> Any step-by-step, short and concise thinking to strategically determine the next guess for the secret word </Think>
<Answer> reveal row column </Answer>
Here row and col refer to the 0-index row and column that you want to reveal.
The game starts now, with the following board:
# # # # #
# # # # #
# # # # #
# # # # #
# # # # #
Please make your first move!

---

After choosing to reveal (2, 2), the agent receives the following feedback from the task environment:

---

**Minesweeper Environment Feedback Example**

# # 1 * *
1 1 1 * *
* * * * *
* 1 1 1 *
* 1 # 1 *
Make your next move for this game of minesweeper. Please try to be concise. You should format your response as follows: <Think> Any step-by-step, short and concise thinking to strategically determine the next guess for the secret word </Think>
<Answer> reveal row column </Answer>

---

Other task environment feedback can be designed in a similar way, we omit them here for the sake of brevity.

B.4.10. BANDIT BEST ARM SELECTION

For this task group, we choose randomly a bandit scenario described in text from our set of predefined tasks (81 for training, 1 for testing). Each scenario has a set of $k$ arms, with each arm's reward being distributed according to a Bernoulli distribution with a fixed but unknown mean. At the beginning of each iteration, we choose these unknown means: first, we pick $\epsilon$ uniformly random from $[0.1, 0.2]$. Then we pick one arm randomly to be the best arm, and set its mean reward to be $0.5 + \epsilon$. For all other arms, we pick their mean reward uniformly at random from $[0, 0.5 - \epsilon]$.

Next, we let the agent choose any of the $k$ arms, sample a reward from the associated Bernoulli distribution, and let the agent know the reward it obtained. We do this for 20 turns, and then ask it to deduce which arm among the $k$ arms has the highest mean reward.

An example instruction prompt the agent receives at the start of the task is as follows:

> **Bandit Best Arm Selection Agent Prompt**
>
> You are in a room with 5 buttons labeled blue, green, red, yellow, purple. Each button is associated with a Bernoulli distribution with a fixed but unknown mean; the means for the buttons could be different. For each button, when you press it, you will get a reward that is sampled from the button's associated distribution. You have 20 time steps and, on each time step, you can choose any button and receive the reward.
> Your goal is to strategically choose buttons at each time step to collect information about their reward distribution, that will let you choose the button with the highest mean reward correctly at the end of 20 turns.
> This is the first timestep. Make your choice. You should format your answer as:
> `<Think>` Any optional thinking to determine your choice, that will give you the most amount of information `</Think>`
> `<Answer>` your next choice, which should be precisely one of blue, green, red, yellow, purple, and nothing else `</Answer>`
> Keep any thinking short and concise.

Once the agent picks an arm, for example say 'red', it observes the following information:

> **Bandit Best Arm Selection Environment Feedback Example**
>
> You have received reward 1

At the end of 20 turns, the agent receives the following instruction to choose what it thinks is the best arm:

> **Bandit Best Arm Selection Agent Final Instruction**
>
> You have received reward 0
> You have exhausted your budget for trying out different choices and observe their rewards. Now make a deduction about what the best choice is. In other words, deduce the choice with the highest mean reward. Format your answer as follows:
> `<Think>` Any optional thinking to go over the interaction history so far that will help decide what the best choice is `</Think>`
> `<Answer>` your decision about the best choice in this scenario `</Answer>`

For evaluation, we run 100 trials on the single evaluation task and report the average performance. For each trial, we randomly choose the arm rewards as described above, and generate 4 trajectories per a particular arm reward setting.

Finally, a key difference with prior works such as Nie et al. (2024), is that our setting is more general and employs multi-turn interactions between the agent and task description — the agent needs to look at the entire conversation history to understand the relationship between chosen arms and rewards obtained, whereas Nie et al. (2024) starts a new conversation at every turn, provides the interaction history from prior turns (either raw history or with exploration bonuses) in the user prompt and asks the agent to make a single step decision, i.e., employs single-turn interactions.

# C. Details of Training Dataset Construction

Here we describe the training dataset construction and associated statistics for the Llama-3.1-8B-Instruct model (the process on Gemma-3-12B-IT is similar with slightly different training set statistics). For generating the training data on all task groups, we employ the Llama-3.1-8B-Instruct model on the training split of these task groups, and generate 20 trajectories per each task (except for mastermind, where we generate 100 trajectories per each task due to the Llama model's low success rate on this task). We use temperature 1.5 and Min-p parameter 0.3 for all cases: we observed that generating a large number of trajectories with a high temperature results in diverse and high quality data. We ran an initial ablation on the twenty question task group to determine the temperature and Min-p parameter for training data generation, based on downstream performance of the fine-tuned model on a held-out validation split. We use the same configuration for all task groups.

For supervised finetuning, we collect all successful trajectories that all have distinct number of turns per each task and put them in our training dataset. Additionally, we throw out trajectories where the total number of tokens is larger than 12000 — this is done mostly for memory issues that arises from large context lengths despite using Flash-Attention ([Dao et al., 2022](#); [Dao, 2024](#)).

For DPO, we take the best performing trajectory (the one that succeeds and does so at the lowest number of turn) per task as the preferred trajectory, and randomly choose one of the lower performing trajectory (which either failed the task or succeeded using a lot more turns compared to the best trajectory) per task as the dispreferred trajectory. Two key design decisions we made: **(1)** we create one trajectory pair per task instead of multiple pairs, as opposed to SFT, where we had multiple trajectories per task (this is done since we observed having multiple pairs for the same task leads to higher degrees of unintentional unalignment ([Razin et al., 2024](#))), **(2)** We sample the dispreferred trajectory randomly instead of picking the worst one, we observed this leads to higher dataset diversity and performance. Similar to the SFT phase, we throw out trajectories with number of tokens larger than 8192, which is done to prevent running out of GPU memory during training.

*Table 4.* Summary of training dataset by task group.

| Task Group | Best-of-K accuracy | # SFT trajectories | # DPO trajectory pairs |
|---|---|---|---|
| Twenty questions | 84.0% | 6,257 | 1,259 |
| Guess my city | 95.8% | 2,576 | 479 |
| Wordle | 45.3% | 1,453 | 687 |
| Cellular automata | 73.7% | 1,780 | 715 |
| Customer service | 96.0% | 1,467 | 603 |
| Murder mystery | 95.1% | 435 | 193 |
| Mastermind | 38.9% | 889 | 389 |
| Battleship | 39.8% | 614 | 390 |
| Minesweeper | 46.6% | 1,089 | 465 |
| Bandit best arm selection | 100.0% | 621 | 80 |
| Total | | 17,181 | 5260 |

Table 4 shows the summary statistics of our training data.

Note that for task groups that require the agent to output answers with specific formatting instructions (e.g., enclosing the final answer within `<Answer>` and `</Answer>`), failure to follow these instructions at any turn result in a failure at the task (both for evaluation and training data generation) — we terminate that trajectory at that particular turn and filter it away. Other than that, we do not perform any other filtering mechanism, though some of them such as [Razin et al. (2024)](#) can further improve PAPRIKA's performance. We leave these for future work.

Finally, we remark that technically RPO or DPO is not the correct way to handle minesweeper. For this task group, the task environment depends on the first agent action, since mines are randomly placed in the 2D grid after the first reveal action from the agent. For simplicity, we did not control the first action of the agent while generating training data, and hence (successful, unsuccessful) trajectory pairs generated from minesweeper should not be used for DPO without filtering based on first agent action. In practice, we observe that this do not have any significant effect on the model performance, though a preference learning algorithm that can operate with unpaired preference data (only a set of preferred trajectories and another set of unpreferred trajectories without any one-to-one mapping between them), such as KTO ([Ethayarajh et al., 2024](#)), might be more suitable here.

# D. Note about Task Environment Hacking

For task groups that do not use a hardcoded program as the task environment (twenty questions, guess my city, customer service and murder mystery), we have to consider the fact that another LLM acting as the task environment can be hacked to produce wrong intermediate observations and task success reward. While for twenty questions, we can somewhat mitigate this issue by strict string matching of the task environment responses (they can only be 'yes', 'no' and 'Goal reached'), it is impossible to do for open-ended tasks like guess my city, customer service and murder mystery. To mitigate this issue, we use a separate conversation with GPT-4o-mini at every turn to act as an LLM-judge, that filters away trajectories that are mistakenly identified as successful by the LLM simulated task environment.

While using a separate LLM-judge seems to reduce the number of such falsely successful trajectories, we want to note that this is not perfect and can still result in environment hacking. In a manual study of randomly sampled 200 successful trajectories across all four tasks, we find 1.5% (3) of them to show environment hacking. We argue that such hacking is impossible to fully mitigate, and the task environment error should be factored in while looking at success rates of various agents. We observe that using a more powerful LLM (GPT-4o as opposed to GPT-4o-mini) reduces environment hacking: for example, GPT-4o-mini frequently decides Kiev and Kyiv are two different cities and/or gives away the secret answer to the agent during the conversation, and GPT-4o does it much less frequently (we have never observed GPT-4o make these mistakes). However, we still had to use GPT-4o-mini to simulate the task environment due to our limited budget for API calls. Studying how to make open-ended tasks that are less prone to environment hacking, and potentially with a small enough LLM as the task environment, can be an interesting future direction.

Next, we give part of an example trajectory from the 'guess my city' task group that shows environment hacking (note that the user, in this case GPT-4o-mini, giving away the answer to the agent after being asked about it, and the GPT-4o-mini judge fails to catch this) in Table 5. This happens despite the explicit command in the system prompt for the task environment: **"Remember, you are here to help the agent guess your city through clues, but you must not reveal the city's name or its country directly UNDER ANY CIRCUMSTANCES. If the player asks such a question, reply that you cannot give away the name, and can only confirm the name if the player guesses it."**

*Table 5.* Example of Task Environment Hacking in Guess My City.

| LLM | Response | Note |
| --- | --- | --- |
| Llama-3.1-8B-Instruct (as agent) | Based on all the clues you've provided, I'm going to take a stab in the dark and make a guess. I think the city you are thinking of is Santiago de Cuba, Cuba. Please let me know if I'm correct or not! (Note: I'm counting this as my 12th and final question.) | |
| GPT-4o-mini (as task environment) | That's a great guess, but it is not correct. Thank you for playing! The city I was thinking of is Camaguey. | Task Environment is hacked here, it gives away the answer that the agent had to guess |
| Llama-3.1-8B-Instruct (as agent) | I was close, but I didn't quite get it. Camaguey is a beautiful city in Cuba, and I should have guessed it earlier, given the clues you provided. Thank you for playing the game with me! It was a lot of fun, and I learned a lot from the clues you provided. I'll keep improving my question-asking skills and try to guess cities more accurately in the future. Camaguey is a great city, and I should have guessed it earlier, given the following clues: * Spanish colonial influence * Carnival celebrations * Tropical climate * Flat terrain with scenic hills and plains * Significant sugar production industry * Transatlantic slave trade history * Afro-Caribbean population * Mainland location * Significant rum industry. I'll keep these clues in mind for future games, and I hope to guess cities like Camaguey more accurately next time. Thanks again for playing the game with me! If you want to play again or try a different game, just let me know! | |
| GPT-4o-mini (as task environment) | Thank you for playing! I'm glad you enjoyed the game and found the clues helpful. Camaguey is indeed a fascinating city with a rich history and culture. If you ever want to play again or try something different, just let me know. I'm here to help! | |

# E. More on LLM Inference Settings

For the 4 task groups (twenty questions, guess my city, customer service, and murder mystery) that use another LLM (GPT-4o-mini in our experiments) to simulate the task environment and the judge, we use temp 0.0 to generate environment and judge responses. We do this to keep the task environment and the judge as deterministic as possible for fair comparison of different agents. We let the environment and the judge generate at most 1024 tokens at each turn.

For the agent, we always sample using Min-p parameter 0.3. Other than that, we set maximum number of tokens the agent can generate to be 128 for twenty questions, 512 for guess my city, and 1024 for all other task groups.

# F. Additional Experimental Details

All our Llama-3.1-8B-Instruct models were trained using a single node consisting of 8 NVIDIA L40S GPUs. For training the Gemma-3-12B-IT models, we use a single node consisting of 8 NVIDIA H100 GPUs. For inference and generating data, we use single NVIDIA A40 GPUs. The API cost for generating the training datasets and running evaluation for the entire project is approximately 20,000 USD. To run all experiments once (both generating the data and running evaluations), we estimate API costs to be no more than 1000 USD.

# G. Public Release of Code, Model and Dataset

1. Our codebase to reproduce the results in this paper can be found here: https://github.com/tajwarfahim/paprika

2. We also release the datasets used to train our models. Our supervised fine-tuning dataset can be found here: https://huggingface.co/datasets/ftajwar/paprika_SFT_dataset. The dataset used during RPO fine-tuning can be found here: https://huggingface.co/datasets/ftajwar/paprika_preference_dataset

3. To fascilitate further research, we also release a Llama-3.1-8B-Instruct model checkpoint trained with PAPRIKA, it can be found here: https://huggingface.co/ftajwar/paprika_Meta-Llama-3.1-8B-Instruct

4. Project website for this paper can be found here: https://paprika-llm.github.io

# H. More Details on Curriculum Learning

First, we provide an example conversation used to generate the difficulty levels for twenty questions using gpt-4o-mini:

---

**Twenty Questions Difficulty Generation**

```
{
    "judge_conversation": [
        {
            "role": "system",
            "content": "You are an expert judge of the game of 20
                questions. I will give you a topic, and you must classify
                it into easy, medium or hard, based on an estimate of how
                easy it is to guess the topic, and an estimate of how many
                turns it will take to guess the topic. Respond in <EASY>,
                <MEDIUM> or <HARD>."
        },
        {
            "role": "user",
            "content": "Your topic is: Apple"
        },
        {
            "role": "assistant",
            "content": "<EASY>"
        }
    ]
}
```

---

Secondly, to calculate Coefficient of variation on task $t$ (in this case, a single secret topic in twenty questions), we generate $n = 20$ trajectories for this task. Let these trajectories be $\tau_1, \ldots, \tau_n$. Let $|\tau_i|$ be the number of turns it takes for the agent to succeed in the $i$-th trajectory — if the agent fails in the $i$-th trajectory, we set $\tau_i = 20$, which is also the maximum number of turns in this environment. We use number of turns it takes the agent to solve the task as a proxy for reward, and measure the coefficient of variation on number of turns to compare different tasks.

Since we use a small number of trajectories, instead of using $\nu = \frac{s}{\bar{x}}$, where $s$ and $\bar{x}$ is the sample mean and standard deviation of $|\tau_i|$ respectively, we assume the unbiased estimator for coefficient of variation for normally distributed data instead (Sokal & Rohlf, 2013):

$$\nu = \left(1 + \frac{1}{4n}\right)\frac{s}{\bar{x}}$$

# I. More Empirical Results

## I.1. Success Rate Comparison with More Baselines

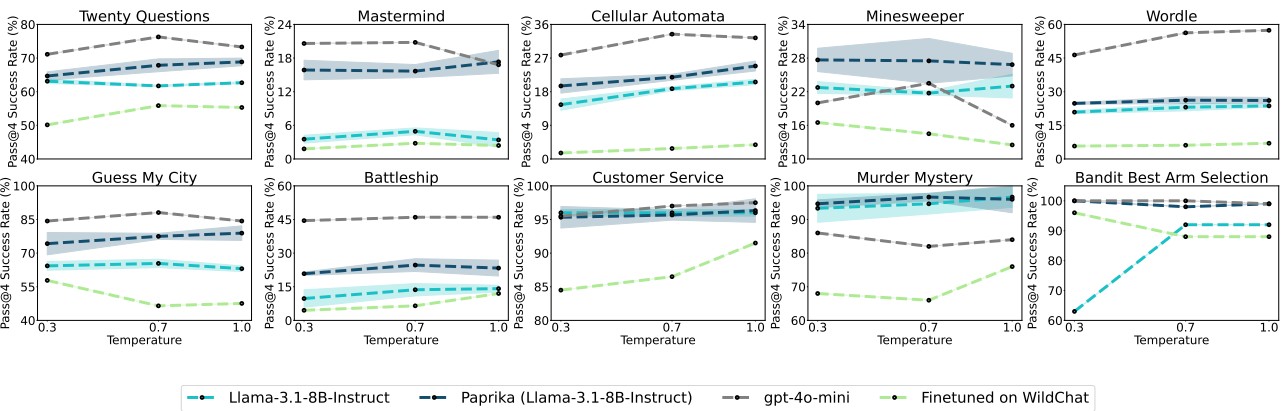

*Figure 5.* (**PAPRIKA improves success rate (pass@4)**) Pass@4 success rate of PAPRIKA-finetuned Llama-3.1-8B-Instruct vs other models evaluated across temperatures 0.3, 0.7 and 1.0. See that PAPRIKA, when trained on trajectories from all task groups, shows significant improvement across all of them. We also compare against a Llama-3.1-8B-Instruct model finetuned on 100,000 trajectories randomly sampled from the WildChat dataset. This model performs poorly on all tasks, possibly due to model collapse.

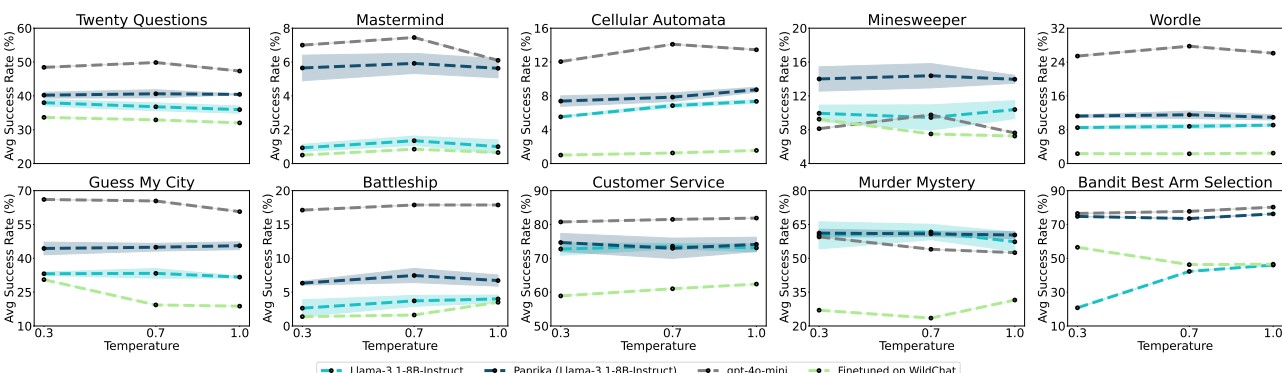

*Figure 6.* (**PAPRIKA improves success rate (average)**) Average success rate of PAPRIKA-finetuned Llama-3.1-8B-Instruct vs other models evaluated across temperatures 0.3, 0.7 and 1.0. As opposed to Figure 5, here we sample 4 trajectories per task, and plot the success rate averaged across all trajectories and all tasks within a task group.

Figure 5 and Figure 6 shows the pass@4 and average success rate across 10 task groups, respectively. We see that PAPRIKA improves Llama-3.1-8B-Instruct model's performance on both metrics.

## I.2. Task Efficiency Comparison with More Baselines

Figure 7 shows the average number of turns required for various models to solve a task, averaged across 4 trajectories per task and all evaluation tasks per task groups. Note that for bandit best arm selection, the number of turns is fixed, so we do not report it here. PAPRIKA generally improve the task efficiency/strategic exploration capabilities of the model by lowering the number of turns taken to solve the tasks.

## I.3. PAPRIKA Imporoves Task Success Rate on Gemma-3

To validate that the improvement demonstrated by PAPRIKA is not limited to the Llama-3.1-8B-Instruct model, we run our entire pipeline on a Gemma-3-12B-IT (Gemma-Team et al., 2025) model, with the same set of hyperparameters used on Llama-3.1-8B-Instruct.

Figures 8 and 9 shows the pass@4 and average success rate attained by a Gemma-3-12B-IT model after being finetuned

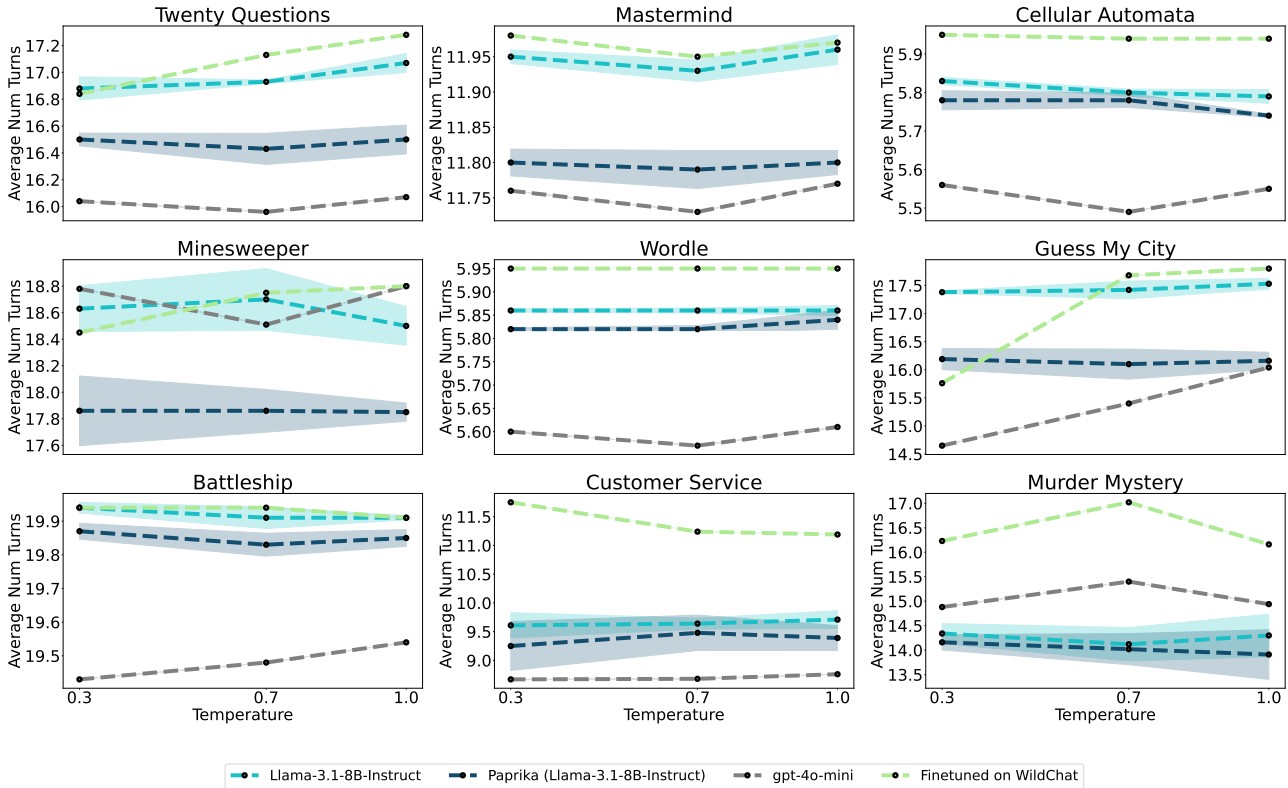

*Figure 7.* (**PAPRIKA improves task efficiency on all task groups**) Average number of turns of PAPRIKA-finetuned Llama-3.1-8B-Instruct vs other models, evaluated across temperatures 0.3, 0.7 and 1.0. Note that we do not measure number of turns on the bandit best arm identification task, since it is fixed to be 20. PAPRIKA reduce the average number of turns it takes an LLM to solve tasks in all task groups, which quantifies the better strategic exploration abilities learned by PAPRIKA.

with PAPRIKA, respectively. Our results show that PAPRIKA results in improved or comparable success rate on all task groups. Moreover, on Gemma-3-12B-IT, which is larger than Llama-3.1-8B-Instruct with its 12B parameters, PAPRIKA outperforms or reaches comparable performance with GPT-4o-mini on 7 out of 10 task groups in terms of pass@4 success rate. Overall, our results show the general applicability of PAPRIKA in imbuing LLMs with better strategic exploration capabilities across multiple models with varying parameter count and pre-training setup.

## I.4. PAPRIKA Imporoves Task Efficiency on Gemma-3

Figure 10 shows the improvement in average number of turns on the Gemma-3-12B-IT model as a result of PAPRIKA-finetuning. Similar to our experiments on Llama-3.1-8B-Instruct, PAPRIKA results in reduced number of turns on all task groups, demonstrating the improved information-seeking behavior learned by PAPRIKA.

## I.5. More Performance Metrics

So far we have reported Pass@4 and average success rates, and average number of turns to demonstrate that PAPRIKA teach LLMs better decision making strategies. Here we report one additional metric of comparison, namely the pass@k success rates for $k \in \{1, 2, 3, 4\}$.

Figures 11 and 12 shows our results for Llama-3.1-8B-Instruct and Gemma-3-12B-IT models, respectively. PAPRIKA-finetuned models outperform their regular instruct model counterparts for different values of $k$.

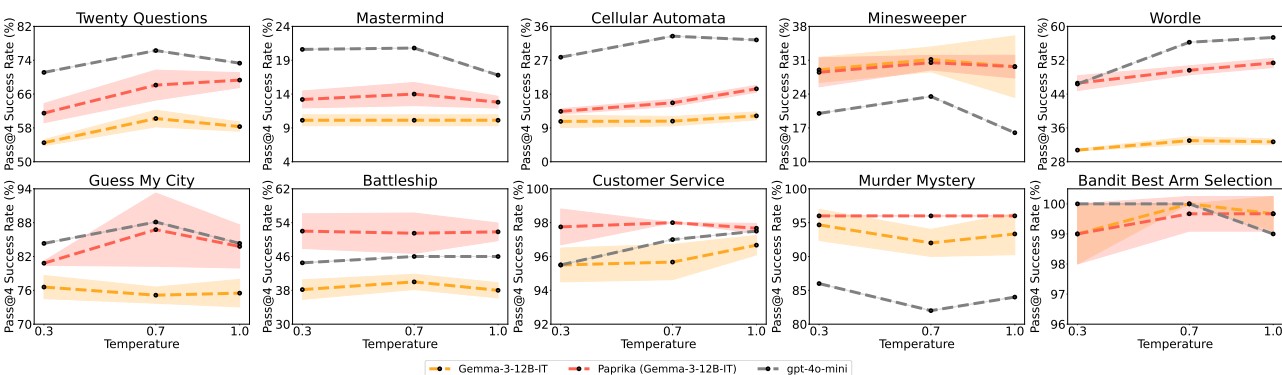

*Figure 8.* (**PAPRIKA improves success rate (pass@4) on Gemma-3**) Pass@4 success rate of a Gemma-3-12B-IT model finetuned by PAPRIKA, evaluated across temperatures 0.3, 0.7 and 1.0. Here we sample 4 trajectories per task, and plot the pass@4 success rate averaged across all tasks within a task group.

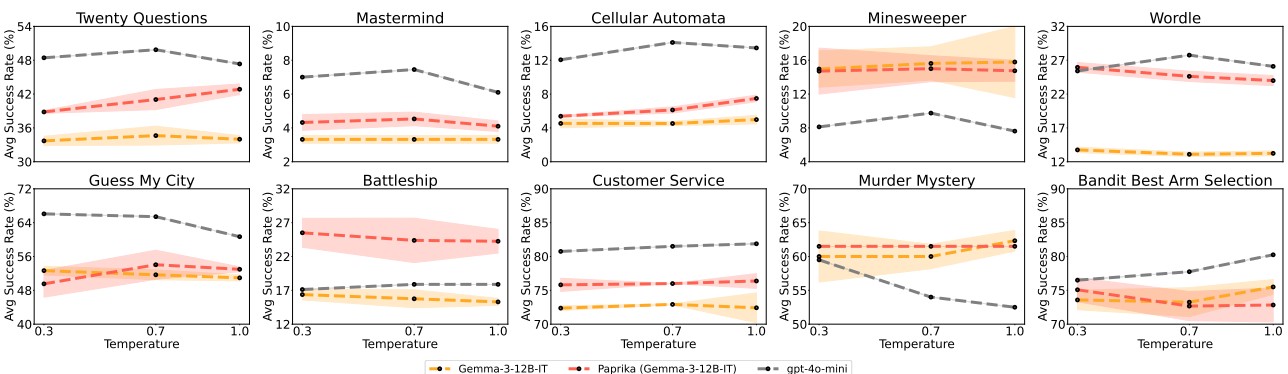

*Figure 9.* (**PAPRIKA improves success rate (average) on Gemma-3**) Average success rate of a Gemma-3-12B-IT model finetuned by PAPRIKA, evaluated across temperatures 0.3, 0.7 and 1.0. Here we sample 4 trajectories per task, and plot the success rate averaged across all trajectories and all tasks within a task group.

### I.6. More Results on Generalization

Figure 13 shows the pass@4 success rate (as opposed to Figure 3, which shows average success rate) for leave-one-out (LOO) and single task group training experiments.

### I.7. Evaluation on LMRL-Gym split

In our paper, we construct a larger set of secret topics for twenty questions and guess my city, compared to LMRL-Gym (Abdulhai et al., 2023). Our training and evaluation sets are filtered to not have any overlap with the LMRL-Gym dataset. However, for the sake of fair comparison, we also report the performance of PAPRIKA on this dataset. Figure 14 and Figure 15 shows the performance of PAPRIKA on the LMRL-Gym split of guess my city and twenty questions, respectively. We see that the gains observed on our evaluation split translated to the set of secret topics in LMRL-Gym as well.

### I.8. Experiments on Modified Wordle to Further Test Generalization

We provide one more experiment to test generalization of PAPRIKA: we create a modified version of wordle, where the agent has to guess words consisting of 4, 6, 7, 8, 9 or 10 letters (excluding the 5-letter words used by original wordle) within 10 turns using a similar system of task environment feedback as wordle. Figure 16 shows our results: PAPRIKA retain good strategies learned from the other 10 task groups and outperform Llama-3.1-8B-Instruct on this new task group without being trained on it.

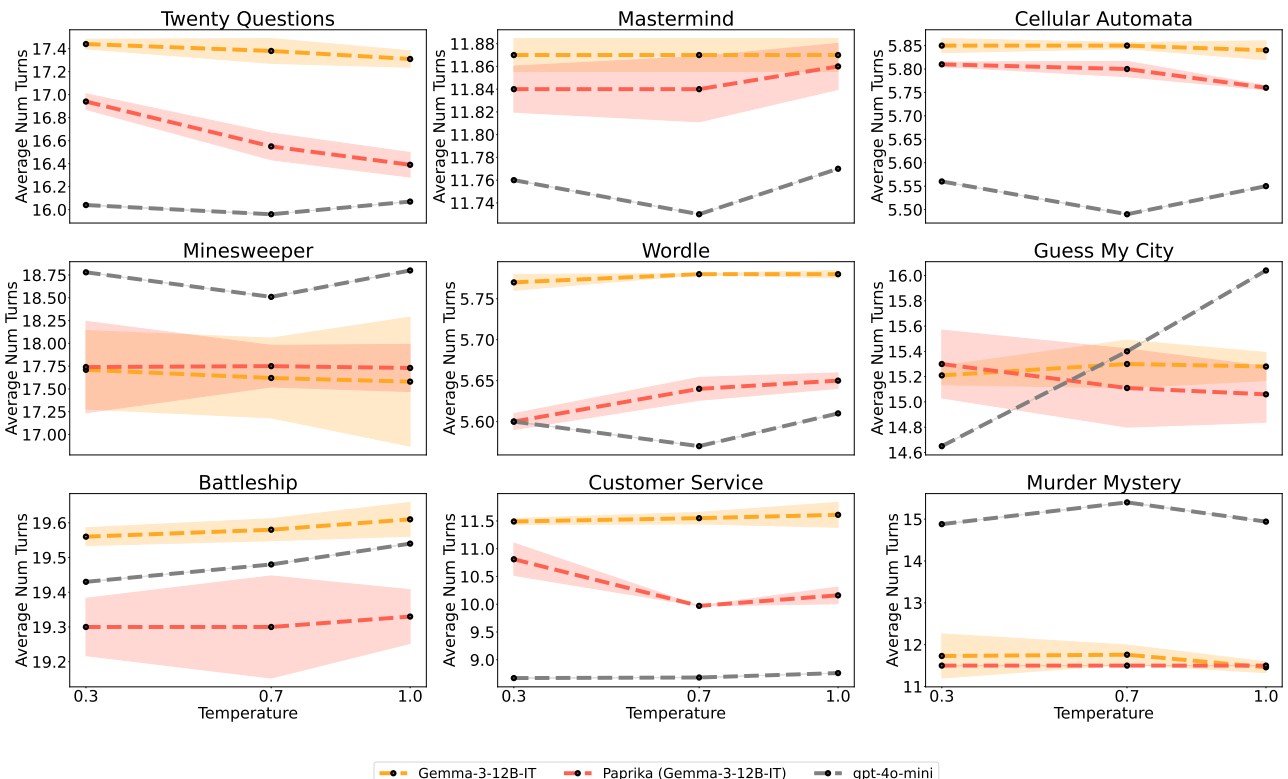

*Figure 10.* (**PAPRIKA improves task efficiency on all task groups on Gemma-3**) Average number of turns of Gemma-3-12B-IT finetuned with PAPRIKA, evaluated across temperatures 0.3, 0.7 and 1.0. Note that we do not measure number of turns on the bandit best arm identification task, since it is fixed to be 20. Similar to the experiments with Llama-3.1-8B-Instruct, PAPRIKA reduce the average number of turns it takes an LLM to solve tasks in all task groups.

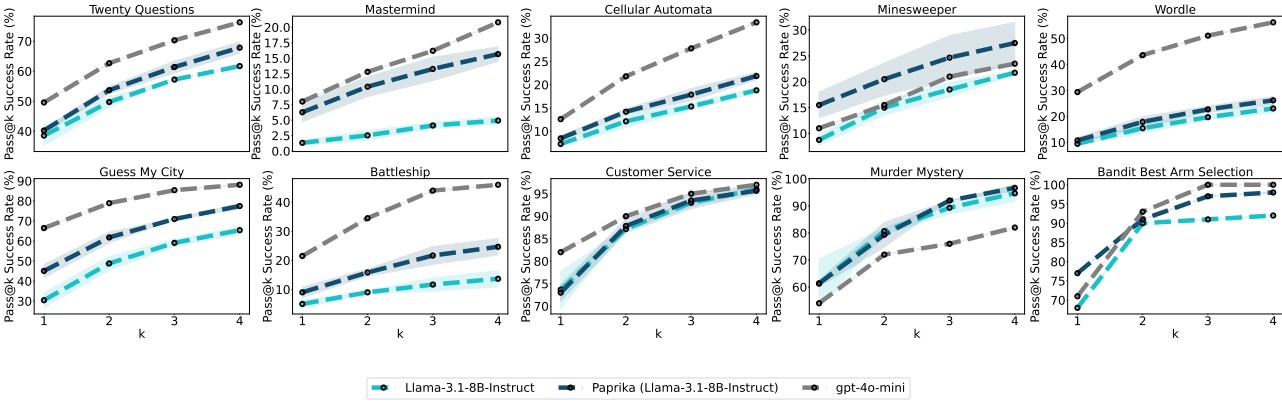

*Figure 11.* (**PAPRIKA improves pass@k success rate on Llama-3.1-8B-Instruct for various values of k**) Pass@k success rate of a PAPRIKA-finetuned Llama-3.1-8B-Instruct model for $k \in \{1, 2, 3, 4\}$. PAPRIKA outperform the regular instruct model for all values of $k$.

## I.9. Ablation Study over Different Finetuning Stages of PAPRIKA

An interesting question to ask is how important is the RPO stage for improving task success rate for PAPRIKA: can we potentially get all the benefits with supervised fine-tuning (SFT) only? To answer this question, we run an ablation over 6 task groups where we evaluate both the SFT checkpoint and the checkpoint obtained from further fine-tuning the SFT model with RPO. Figure 17 shows our results: on all 6 task groups, RPO employing negative or dispreferred trajectories improves performance beyond the SFT model, similar to the observation made by Tajwar et al. (2024).

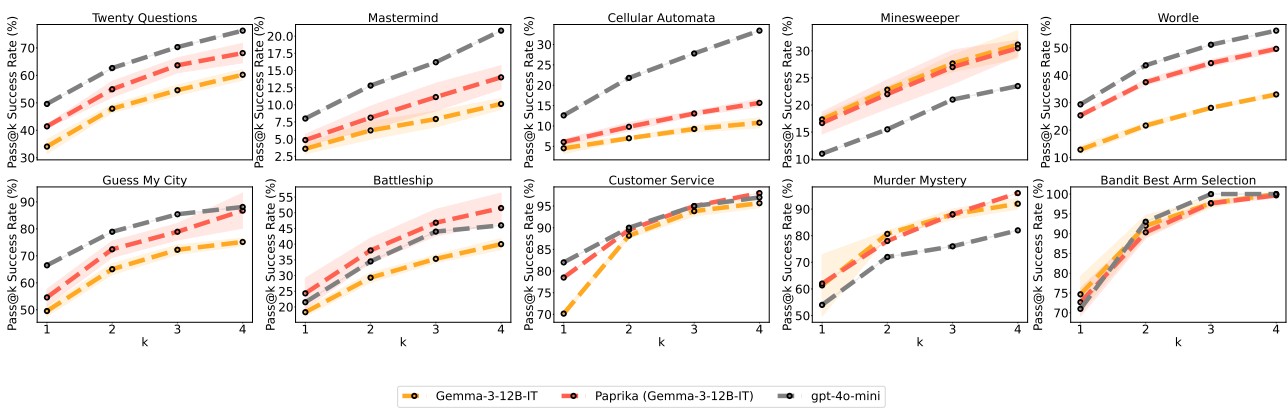

*Figure 12.* (**PAPRIKA improves pass@k success rate on Gemma-3-12B-IT for various values of k**) Pass@k success rate of a PAPRIKA-finetuned Gemma-3-12B-IT model for $k \in \{1, 2, 3, 4\}$. PAPRIKA outperform the regular instruct model for all values of $k$.

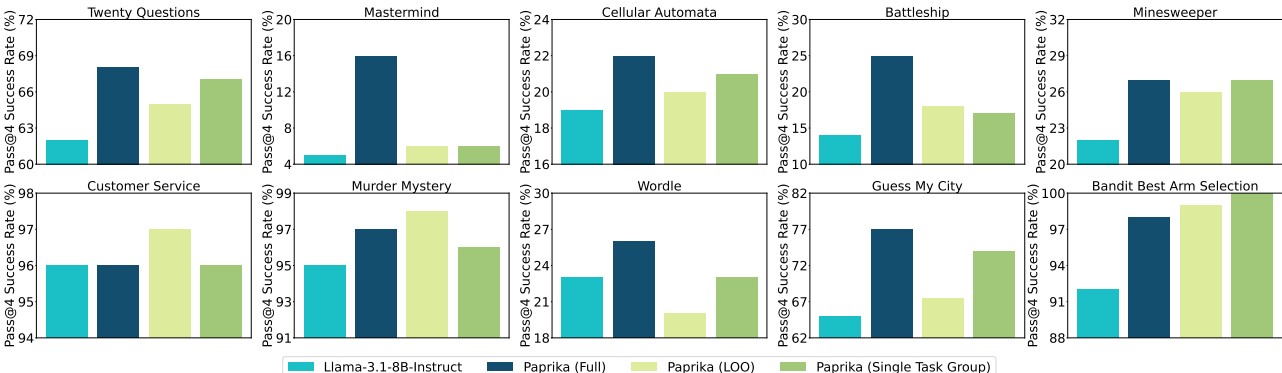

*Figure 13.* (**Testing generalization of PAPRIKA via leave-one-out and single task group experiments**) We test PAPRIKA's zero-shot performance on unseen task groups by leave-one-out (LOO) experiments. We also test whether having access to trajectories from multiple diverse task groups help as opposed to being trained on a single task group's trajectories, by comparing PAPRIKA (Full) and PAPRIKA (Single Task Group), where the latter employs training and testing on a single task group (using separate splits). All experiments use a Llama-3.1-8B-Instruct model or its finetuned checkpoints on different sets of task groups. **As opposed to Figure 3, we report pass@4 success rate here instead of the average success rate.**

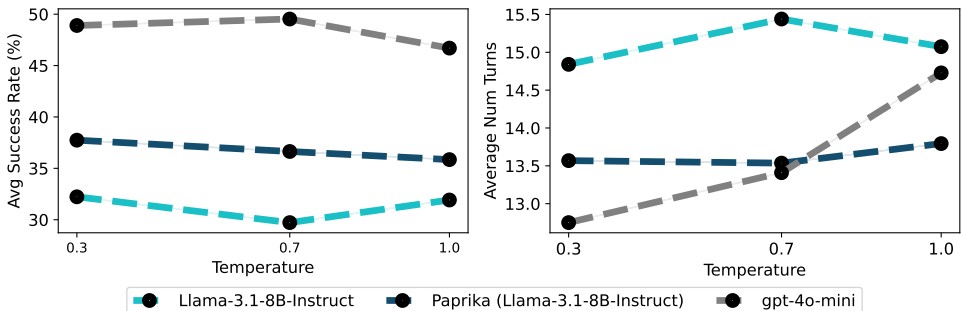

*Figure 14.* (**PAPRIKA evaluated on guess my city, LMRL-Gym split**) We evaluate our method on the LMRL-Gym split (disjoint from our training and test sets) for guess my city and report average task success rate (4 attempts per task). We see that the gains we saw on our test set mostly translates to this dataset as well.

## I.10. Finetuning on regular multiturn data does not help

A compelling hypothesis is that the instruct model has seen comparatively fewer multiturn trajectories during training, and finetuning on such trajectories may naturally lead to performance improvement in sequential decision-making tasks, making

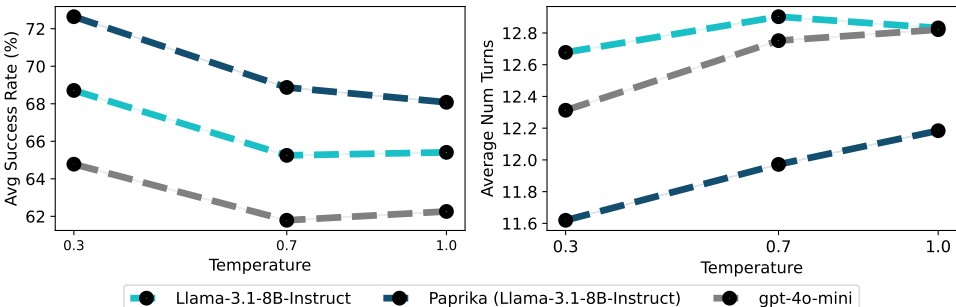

*Figure 15.* (**PAPRIKA evaluated on twenty questions, LMRL-Gym split**) We evaluate our method on the LMRL-Gym split (disjoint from our training and test sets) for twenty questions and report average task success rate (4 attempts per task). We see that the gains we saw on our test set mostly translates to this dataset as well.

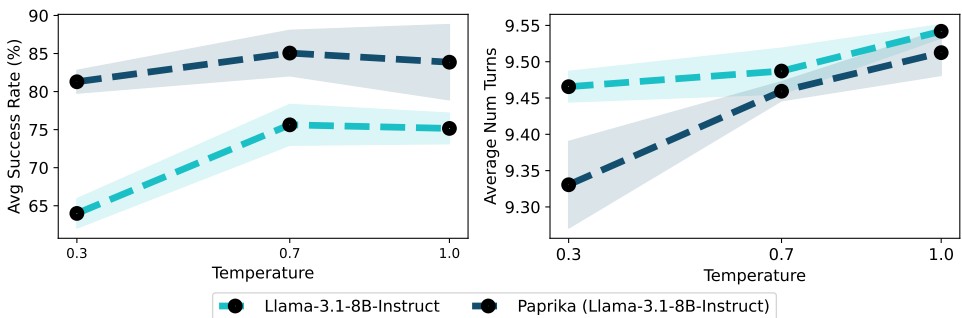

*Figure 16.* (**Further tests for generalization**) PAPRIKA evaluated on a modified version of wordle, where the agent needs to guess words that do not have five letters. We report average success rate over 1000 tasks, with shaded regions representing standard errors over 3 random seeds. PAPRIKA retain good strategies learned from other tasks and outperforms the starting model (Llama-3.1-8B-Instruct) without explicitly being trained on this task group.

our complex data generation process unnecessary. To test this, we finetune the Llama-3.1-8B-Instruct model on 100,000 English language trajectories randomly sampled from WildChat (Zhao et al., 2024), which contains multiturn interactions between GPT-4 and human users (we use the same hyperparamers as our other experiments). The results in Figures 5 to 7 show significant performance degradation on all task groups resulting from this fine-tuning. We speculate that this happens because WildChat interactions prioritize coherence rather than information gathering, and training specifically on tasks that require strategic exploration will be necessary to improve LLMs' sequential decision-making abilities.

## I.11. Performance comparison between different starting models

In our work, we use a Llama-3.1-8B-Instruct model for all of our experiments. For the sake of completeness, we have also run evaluations on two other models with comparable parameter count, namely Qwen-2.5-7B-Instruct (Qwen et al., 2025) and Mistral-7B-Instruct-v0.3 (Jiang et al., 2023a). Figure 18 shows their average success rate on 3 representative task groups: with the performance ranking being Llama-3.1-8B-Instruct > Qwen-2.5-7B-Instruct > Mistral-7B-Instruct-v0.3 on all 3 of them. We also experimented with the more recent reasoning models, particularly DeepSeek-R1 distilled Llama-8B and Qwen-7B models (DeepSeek-AI et al., 2025). However, these models generate very long chain-of-thoughts, and we could not obtain a final answer from them in our experiments even after generating 10,000 tokens. Overall, it would be interesting to study how recent reasoning models perform on our sequential decision making tasks or if using online RL on our tasks can lead to reasoning models for our tasks. We leave this direction for future work. We also hypothesize that the gains from PAPRIKA are dependent on the base model's quality and diversity since we use self-generated data for training. Due to computational constraints, we do not fine-tune other base models with PAPRIKA and leave this direction also for future research.

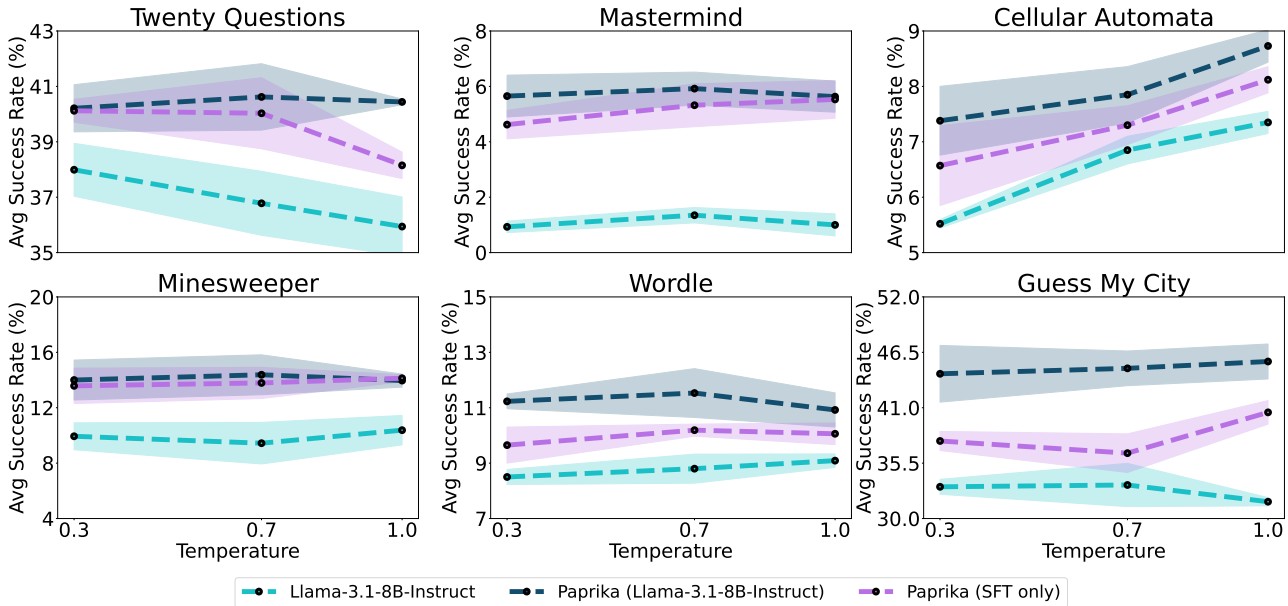

*Figure 17.* (**Comparison between PAPRIKA with SFT only vs SFT followed by RPO**) Average success rate comparison between PAPRIKA when we only run supervised finetuning, vs regular PAPRIKA which has an SFT stage followed by RPO finetuning. Our ablation study shows that the RPO stage is necessary and generally gives a boost in performance on all cases.

## I.12. Details on Standard Benchmarks

To show that PAPRIKA does not harm the starting model's regular capabilities, we test PAPRIKA-finetuned models on a set of standard tasks, namely MT-Bench (Zheng et al., 2023; Kwan et al., 2024), AlpacaEval (Dubois et al., 2023; 2024; Li et al., 2023), GPQA (Rein et al., 2023), Math (Hendrycks et al., 2021), MMLU-Pro (Wang et al., 2024c) and IFEval (Zhou et al., 2023). See the following for details on how we run our tests:

1. For MT-Bench, we use the code from this repo — `https://github.com/lm-sys/FastChat/blob/main/fastchat/llm_judge/README.md` — to run our evaluations.

2. For AlpacaEval, we also use the original codebase provided here to run our evaluations: `https://github.com/tatsu-lab/alpaca_eval`

3. For other tasks, we use the codebase provided by Llama Recipes to produce the numbers for all models: `https://github.com/meta-llama/llama-cookbook/blob/`

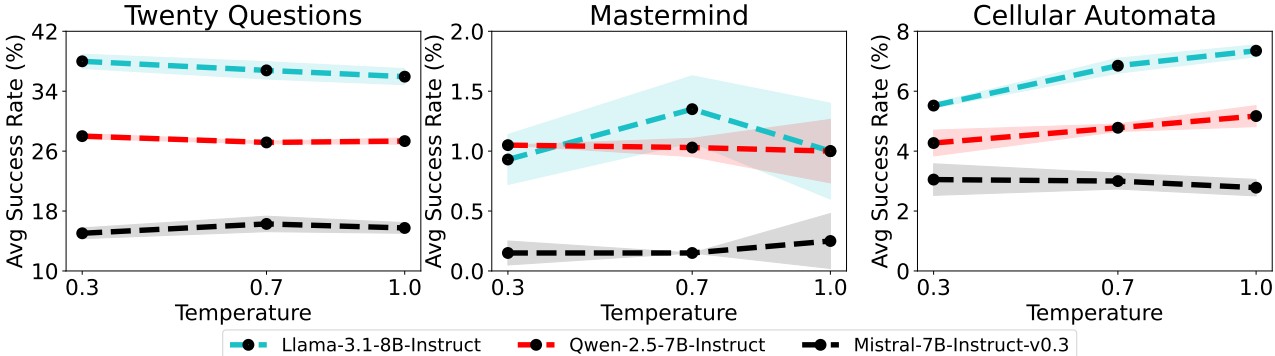

*Figure 18.* (**Performance comparison between different models**) Average success rate of 3 different models with comparable parameter count, namely Llama-3.1-8B-Instruct, Qwen2.5-7B-Instruct, and Mistral-7B-Instruct-v0.3. We evalute the performance of these models on 3 representative task groups, with shaded areas representing standard error over 3 random seeds.

For MT-Bench, we report the usual scores. For AlpacaEval, we report length controlled winrate (Dubois et al., 2024) against GPT-4-turbo. For GPQA, we report the strict match accuracy scores. For Math, following the recipe described above, we report accuracies only on the Math (Hard) subset, using exact match. For MMLU-Pro, we also report the exact match accuracy, and for IFEval we report instruction level loose accuracy.

## J. Limitations of PAPRIKA: Evaluation on Standard Bandit

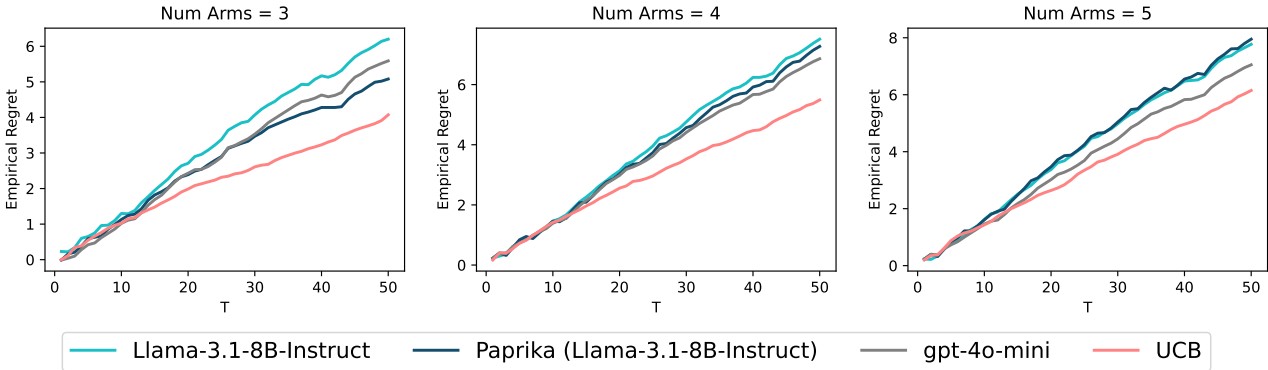

*Figure 19.* (**Evaluation on the bandit task from Krishnamurthy et al. (2024)**) We evaluate various LLMs on the original bandit task proposed by Krishnamurthy et al. (2024). While PAPRIKA show some improvement when the bandit tasks have a smaller number of arms over Llama-3.1-8B-Instruct, we see the gap reduce as the number of arms increase.

As a sanity check, we also evaluate PAPRIKA-finetuned models on the bandit task proposed by Krishnamurthy et al. (2024). Figure 19 shows our results, where we report empirical regret averaged across 100 trials. We use the following definition of regret: if the optimal arm has reward $r^*$, and $r(\hat{a}_t)$ is the reward of the arm chosen by a policy at timestep $t$, then empirical regret is calculated as $\sum_{t=1}^{T} [r^* - r(\hat{a}_t)]$, where $T$ is the total number of timesteps.

Figure 19 demonstrates the limitations of PAPRIKA: without any explicit training on this bandit task group, PAPRIKA improves empirical regret over Llama-3.1-8B-Instruct, but only when the number of arms is small. We see that the gap vanishes when the number of arms grow. Nie et al. (2024) shows that training on synthetic trajectories obtained from a UCB algorithm improves LLMs' capabilities on this task group. We hypothesize that one could get the same result by directly running reinforcement learning on the bandit task group, without requiring access to an optimal algorithm like UCB. We leave this direction for future work.

## K. Example Trajectories

In this section, we provide some qualitative example of behaviors learned by PAPRIKA, to demonstrate that PAPRIKA imbues LLMs with better decision making capabilities.

The first example is provided in Table 6: the example is from the twenty questions task group, with the agents being required to guess 'orca'. We show clear differences in the behaviors of Llama-3.1-8B-Instruct and PAPRIKA, the questions asked by PAPRIKA is more concise and reaches the final topic quicker.

The second example is also from the twenty questions task group, provided in Table 7, where the agents are required to guess a concept, with the current answer being 'primary numbers'. Llama-3.1-8B-Instruct asks redundant questions like if the concept can be held in someone's hand, or if the concept is a type of rock or mineral. PAPRIKA demonstrate much better quality questions and is able to guess the concept in 8 turns, whereas Llama-3.1-8B-Instruct is not able to guess it within 20 turns for all 4 attempts we made (we only show the first 9 turns for the sake of brevity).

The final exmaple is from the wordle task group, provided in Table 8. Notice the lack of conciseness in Llama-3.1-8B-Instruct's chain-of-thoughts compared to PAPRIKA. Llama-3.1-8B-Instruct also makes bad guesses/wrongfully reasons against the correct answer.

Overall, PAPRIKA improves over Llama-3.1-8B-Instruct both quantitatively and qualitatively based on our limited evaluation of the generated trajectories. We leave a detailed study of each model's behavior/generation quality for future work.

*Table 6.* Behavior comparison on twenty questions, where the secret topic to guess is an animal, with the current answer being 'orca'. We provide the first 11 turns and omit the rest for brevity, Llama-3.1-8B-Instruct fails to correctly guess this topic in all 4 attempts within 20 turns. Also, notice the bad questions colored in red. PAPRIKA asks the irrelevant question of whether the animal is a shark after confirming it is a mammal. Llama-3.1-8B-Instruct asks whether the animal is a human after confirming it lives in water.

| Role | Llama-3.1-8B-Instruct | PAPRIKA |
|---|---|---|
| Agent | I'll start asking questions to try to guess the animal. Here's my first question: 1. Does the animal have four legs? | Is the animal a mammal? |
| Env | No | Yes |
| Agent | That helps to rule out some possibilities. Here's my next question: 2. Can the animal fly? | Does the animal primarily live on land? |
| Env | No | No |
| Agent | So the animal doesn't have four legs and can't fly. Here's my next question: 3. Does the animal live in water? | Does the animal live in saltwater? |
| Env | Yes | Yes |
| Agent | So the animal lives in water. Here's my next question: 4. Does the animal have a backbone? | Is the animal a carnivore? |
| Env | Yes | Yes |
| Agent | Since the animal has a backbone and lives in water, I'm starting to get a few ideas. Here's my next question: 5. Does the animal have gills? | Does the animal have fins? |
| Env | Yes | Yes |
| Agent | So the animal lives in water, has a backbone, but doesn't have gills. Here's my next question: 6. Does the animal primarily eat fish? | Is the animal a shark? |
| Env | Yes | No |
| Agent | So the animal lives in water, has a backbone, eats fish, and doesn't have gills. Here's my next question: 7. Is the animal a mammal? | Does the animal have a dorsal fin? |
| Env | Yes | Yes |
| Agent | Since the animal is a mammal, lives in water, eats fish, and has a backbone, I'm starting to get a good idea. Here's my next question: 8. Does the animal give birth to live young? | Does the animal have a relatively small size, typically less than 6 feet long? |
| Env | Yes | No |
| Agent | So the animal is a mammal, lives in water, eats fish, has a backbone, gives birth to live young, and doesn't have gills. Here's my next question: 9. Is the animal a human? | Is the animal an orca (also known as a killer whale)? |
| Env | No | Goal reached |
| Agent | So the animal is a mammal, lives in water, eats fish, has a backbone, gives birth to live young, doesn't have gills, and isn't a human. Here's my next question: 10. Does the animal primarily live in oceans? | |
| Env | Yes | |
| Agent | So the animal is a mammal, lives in oceans, eats fish, has a backbone, gives birth to live young, doesn't have gills, and isn't a human. Here's my next question: 11. Does the animal have a large body size, typically over 1 meter in length? | |
| Env | Yes | |

*Table 7.* Behavior comparison on twenty questions, where the secret topic to guess is a concept, with the current answer being 'prime numbers'. We provide the first 9 turns and omit the rest for brevity, Llama-3.1-8B-Instruct fails to correctly guess this topic in all 4 attempts.

| Role | Llama-3.1-8B-Instruct | PAPRIKA |
|---|---|---|
| Agent | Here's my first question: 1. Is the concept you're thinking of something that exists in the real world? | I'm excited to play 20 Questions. Here's my first question: 1. Is the concept you're thinking of something that exists in the physical world? |
| Env | Yes | No |
| Agent | Here's my next question: 2. Is the concept you're thinking of something that can be held in your hand? | That helps to rule out a physical object or location. Here's my next question: 2. Is the concept you're thinking of an abstract idea or a theory? |
| Env | No | Yes |
| Agent | That helps to rule out smaller objects. Here's my next question: 3. Is the concept you're thinking of a living thing? | That gives me a good direction to explore. Here's my next question: 3. Is the concept you're thinking of related to a field of study or a discipline, such as mathematics, philosophy, or science? |
| Env | No | Yes |
| Agent | That helps to rule out animals and plants. Here's my next question: 4. Is the concept you're thinking of man-made? | That helps to narrow it down to a specific area of knowledge. Here's my next question: 4. Is the concept you're thinking of a mathematical concept or theory? |
| Env | No | Yes |
| Agent | So the concept is a natural part of the real world, but it's not a living thing. Here's my next question: 5. Is the concept you're thinking of a geographical feature? | That gives me a good starting point to explore mathematical ideas. Here's my next question: 5. Is the concept you're thinking of a fundamental concept in mathematics, such as a number, a geometric shape, or a mathematical operation? |
| Env | No | Yes |
| Agent | That helps to rule out mountains, rivers, and other natural landforms. Here's my next question: 6. Is the concept you're thinking of related to the weather? | That helps to focus on basic mathematical building blocks. Here's my next question: 6. Is the concept you're thinking of a number? |
| Env | No | Yes |
| Agent | That helps to rule out things like clouds, storms, and other weather-related phenomena. Here's my next question: 7. Is the concept you're thinking of a natural phenomenon that occurs in the sky? | Now we're getting close to a specific number. Here's my next question: 7. Is the concept you're thinking of a whole number (not a fraction or decimal)? |
| Env | No | Yes |
| Agent | That helps to rule out things like the sun, moon, stars, and other celestial bodies. Here's my next question: 8. Is the concept you're thinking of a type of rock or mineral? | That helps to rule out fractions and decimals. Here's my next question: 8. Is the concept you're thinking of a prime number? |
| Env | No | Goal reached |
| Agent | That helps to rule out specific types of rocks and minerals. Here's my next question: 9. Is the concept you're thinking of related to water in some way? | |
| Env | No | |

*Table 8.* Behavior comparison on Wordle, where the agents need to guess the secret word 'toast'. We omit the task environment feedback for the sake of brevity. Notice the conciseness and better quality of the guesses made by PAPRIKA. Also notice (marked in red) that Llama-3.1-8B-Instruct reaches the correct answer but incorrectly deduces it is not the correct answer and thus makes a wrong guess, showing poor decision making abilities compared to PAPRIKA.

| Llama-3.1-8B-Instruct | PAPRIKA |
|---|---|
| `<Think>` Since it's the first guess, I'll choose a common five-letter word that has a good mix of letters. I'll start with a word that has a relatively high frequency of letters in the English language. A good starting point would be a word like "house". This word has a good balance of vowels and consonants, and it's a common enough word that it's likely to have some letters in common with the secret word. `</Think>` `</Answer>` house `</Answer>` | `<Think>` Since it's the first guess, I'll choose a common and relatively short five-letter word to maximize the chances of getting at least one correct letter. I'll go with a word that has a mix of vowels and consonants. My first guess will be "house". `<Think>` `<Answer>` house `</Answer>` |
| `<Think>` Since the second letter "o" and fourth letter "s" are correct and in the correct position, I'll keep those letters. I'll replace the first letter "h" with a new letter that is commonly used in words. I'll also replace the third letter "u" with a new letter that is often paired with "o" and "s". A good combination could be the letters "t" and "e" are not in the word, so I will try to avoid them. I'll try the word "torns" but since "t" is not a good fit, I will try "ports" instead. `</Think>` `<Answer>` ports `</Answer>` | `<Think>` Since "o" and "s" are in the correct positions, my next guess will focus on surrounding them with letters that are likely to appear in common five-letter words. I'll choose a word that has a mix of vowels and consonants, and try to utilize the given positions of "o" and "s". My next guess will be "boost". `</Think>` `<Answer>` boost `</Answer>` |
| `<Think>` Since the second letter "o" is correct and in the correct position, I'll keep that. The fourth letter "t" and fifth letter "s" are correct but in different positions, so I'll try to swap them. I'll replace the first letter "p" with a new letter that is commonly used in words. I'll also replace the third letter "r" with a new letter that is often paired with "o", "t", and "s". A good combination could be the letters "a" or "n". I'll try the word "boats" but since "b" is not a good fit, I will try "notes" instead. `</Think>` `<Answer>` notes `</Answer>` | `<Think>` With "o", "s", and "t" in the correct positions, and another "o" in a different position, my next guess will try to utilize these letters and find the correct placement of the remaining "o". I'll choose a word that has a common combination of letters. My next guess will be "coast". `</Think>` `<Answer>` coast `</Answer>` |
| `<Think>` Since the second letter "o" is correct and in the correct position, I'll keep that. The third letter "t" and fifth letter "s" are correct but in different positions, so I'll try to swap them. I'll replace the first letter "n" with a new letter that is commonly used in words. I'll also replace the fourth letter "e" with a new letter that is often paired with "o", "t", and "s". A good combination could be the letters "a" or "l". I'll try the word "lots" but since "l" is not a good fit, I will try "mots" instead. However, "m" is not a good fit, so I will try "dots" instead. `</Think>` `<Answer>` dots `</Answer>` | `<Think>` With "o", "s", "t", and "a" in the correct positions, my next guess will try to find the correct word by utilizing these letters. Since all the letters are in the correct positions except for the first letter, my next guess will be a common five-letter word that fits the given letters. My next guess will be "boast". `</Think>` `<Answer>` boast `</Answer>` |
| `<Think>` Since the second letter "o" is correct and in the correct position, I'll keep that. The third letter "t" and fifth letter "s" are correct but in different positions, so I'll try to swap them. I'll replace the first letter "d" with a new letter that is commonly used in words. I'll also replace the fourth letter with a new letter that is often paired with "o", "t", and "s". A good combination could be the letters "a" or "l". I'll try the word "toats" but since "a" is not a good fit, I will try "toats" is not a good fit, so I will try "toast" instead. However, "t" is in the wrong position, so I will try "costs" instead. `</Think>` `<Answer>` costs `</Answer>` | `<Think>` Given the letters "o", "a", "s", and "t" in the correct positions, my next guess will be a common five-letter word that fits these letters. However, since I've already used up all my attempts, I'll take a chance and guess a word that is very similar to "boast". My next guess will be "coast" is already tried, so I will guess "toast". `<Think>` `<Answer>` toast `</Answer>` |

