# OpenReview forum: "Training a Generally Curious Agent"
_ICML.cc/2025/Conference — ICML 2025 oral_

### Official Review · Reviewer_WvQu · 2025-03-10

**Overall Recommendation:** 4

**Summary:**

This paper proposed a fine-tuning approach named PAPRIKA to improve LLM Agent’s decision-making capabilities, along with a curriculum learning algorithm that improves PAPRIKA’s sampling efficiency.
The experimental results show that the proposed PAPRIKA approach improves the success rate among different tasks by a large margin compared to the original base model. And the curriculum learning algorithm further enhanced the Agent’s decision-making capabilities.

**Claims And Evidence:**

Yes. The claims made is supported by evidence.

**Essential References Not Discussed:**

No.

**Experimental Designs Or Analyses:**

Q1:
In Section4.1 Finetuning on regular multiturn data does not help, you fine-tuned Llama-3.1-8B-Instruct on 100,000 English language trajectories sampled from WildChat, why not the whole dataset?


Q2:
In Section 4, PAPRIKA improves LLM decision making abilities, here written when include trajectories from the bandit task in training set can increase rate to 100%, but there is only one test case in your Bandit best arm selection game. That seems not so convincing, why not some more test case?

**Methods And Evaluation Criteria:**

yes.
The proposed method is evaluated by benchamrk dataset.

**Other Comments Or Suggestions:**

No.

**Other Strengths And Weaknesses:**

Strengths
Originality: The paper presents a framework, PAPRIKA, that combines synthetic trajectory generation, a sequential variant of Direct Preference Optimization (DPO), and curriculum learning. This combination introduces a new way to train LLMs for multi-turn decision-making tasks, which distinguishes it from previous approaches that focus mainly on single-turn interactions.


Weeknesses:
Scalability Concerns: Generating large volumes of synthetic trajectories can be computationally expensive. The paper could benefit from a discussion on how this scalability challenge might be addressed, especially when extending the framework to more complex or real-world environments.

**Questions For Authors:**

No. please refer to above questions.

**Relation To Broader Scientific Literature:**

1. This approach addresses limitations highlighted in prior work—such as poor performance on even simple multi-armed bandit tasks (Krishnamurthy et al., 2024)—by training the model on diverse decision-making tasks that require iterative reasoning.

2. In-context learning allows models to adapt to new tasks from a few examples. PAPRIKA leverages this capability and combines it with reinforcement learning principles, effectively training the model to perform in-context reinforcement learning across multiple rounds of interaction.

3. One of the  notable findings is that PAPRIKA-trained models exhibit strong zero-shot performance on unseen tasks. The demonstrated transferability highlights the paper’s contribution to bridging the gap between task-specific training and broader, more adaptable decision-making capabilities.

**Theoretical Claims:**

Q1:
In Algorithm 1, the pseudocode said that there is an input B for parallel experience collection, but in the pseudocode, there is no evidence show that the collection process is parallel. What is the intent for B.

Q2:
In Section3.3, the SFT objective and RPO objective formulations seem to express the training set for SFT and RPO are the same. But in Experimental Setup, here written 15,744 training trajectories for SFT and 4388 trajectories for RPO, so it seems that they are different training set. If different, there should be an explicit difference in the both objective formulations in Section 3.3.

---

> ### Author Rebuttal · Authors · 2025-04-01
>
> We thank the reviewer for their feedback and positive review.
>
> > Q1: In Algorithm 1, the pseudocode said that there is an input $B$ …
>
> We thank the reviewer for catching this typo. $B$ in the pseudocode has the same purpose as $C$, in the sense that we collect $C$ trajectories in parallel for each of the $T$ tasks (selected by the algorithm). The collection of the $C$ trajectories is parallelized across multiple GPUs. We will edit our paper to reflect this more clearly.
>
> One can in-practice also parallelize the data collection across multiple tasks (beyond trajectories for a single task as we do) with more computational resources. In this case, a batched variant of UCB [1, 6] can be used to update the arm parameters, which we leave for future work.
>
> > Q2: In Section3.3, the SFT objective and RPO objective…
>
> We apologize for the confusion. In section 3.3, we define loss functions over any dataset $D$, and do not differentiate between the dataset used for SFT vs RPO stages. We will update the paper to clarify our notation.
>
> However, we also want to provide some clarifications here: for the SFT phase, we collect **all** successful trajectories for each training task $\tau$. During the RPO stage, for each training task $\tau$, we pick the best trajectory and one of worse-scoring trajectories randomly to form **exactly one** preference pair. We sample 20 trajectories per task, a large number of which can be successful and included in the SFT phase, but we construct at most 1 preference pair per task. The reason we only form one preference pair per task is because we observed that using more than one per task can cause unintentional unalignment [2, 3], and lead to performance degradation.
>
> > Q1: In Section 4.1 Finetuning on regular multiturn data does not help…
>
> Fine-tuning on the full WildChat-1M dataset takes close to 10 days with our computational resources and we could not run that. Moreover, certain trajectories in WildChat have very long context length (> 100,000) and we do not have the resources to train on such long trajectories. Also **note that we only ~20K trajectories for Paprika**, showing that targeted synthetic trajectory generation is much more sample efficient for the tasks we care about in this work.
>
> > Q2: …but there is only one test case in your Bandit best arm selection…
>
> Apologies for the confusion. Even though we have only one test case (e.g., picking colors), **we test our models on 100 iterations of the test case, each time randomly sampling the bandit arm probabilities**. For each specific bandit arm probabilities, we run 4 iterations of the game, and report the average pass@4 success rate to be 100%. We would edit the paper to make this point clearer.
>
> To follow the reviewer’s suggestion, we generate **20 more bandit tasks** (so, 20 problem description + different arm names) using gpt-4o-mini and test Paprika on them, with pass@4 success rate reducing slightly to 98%, still outperforming the regular instruct model (See [this Figure](https://ibb.co/xtnt563Z), first row, first column)
>
> > Weeknesses: Scalability Concerns…
>
> Sampling trajectory is indeed the main time bottleneck. However, we believe this is not too different from other RL tasks. Sampling is the only way we can get experience from the model, and it is still considerably cheaper than collecting human demonstrations. Recent works have found similar approaches to be very effective for other domains like math or coding [4, 5]. Collecting expert data and doing domain specific pre-training or mid-training can improve sample efficiency during the RL phase [7]. Another scalability concern is designing particular tasks for training, and a potential solution is to train another LLM to generate these tasks and then use our curriculum algorithm to select which ones to train on at every step.
>
> ## Additional Experiments
>
> We have also run additional experiments as per the other reviewers’ suggestions:
>
> 1. Paprika with Gemma-3-12B-IT: Our experiments show that Paprika works with Gemma-3-12B-IT as the base model. We also see the **Paprika fine-tuned model being comparable or better than GPT-4o-mini in 7/10 task groups**. See full results [here](https://ibb.co/dsv8xG3q).
>
> 2. More tests for generalization: We extend Figure 4 by running leave-one-out experiments on 5 more task groups, and Paprika (LOO) improves over the instruct model in 9/10 task groups, showing strong generalization: [Figure](https://ibb.co/xtnt563Z)
>
> ## References
>
> [1] Multi-Armed Bandit Problem and Batch UCB Rule
>
> [2] Unintentional Unalignment: Likelihood Displacement in Direct Preference Optimization
>
> [3] Preference Fine-Tuning of LLMs Should Leverage Suboptimal, On-Policy Data
>
> [4] DeepSeek-R1: Incentivizing Reasoning Capability in LLMs via Reinforcement Learning
>
> [5] OpenAI o1 System Card
>
> [6] Perchet, Vianney, et al. "Batched bandit problems." (2016): 660-681
>
> [7] Cognitive behaviors that enable self-improving reasoners, or, four habits of highly effective stars

---

> > ### Comment · Reviewer_WvQu · 2025-04-04
> >
> > Thank you for your detailed response. After reviewing the rebuttal, I have updated the score to Accept.

---

> > > ### Author Response · Authors · 2025-04-04
> > >
> > > Dear Reviewer WvQu,
> > >
> > > Thanks a lot for your thoughtful questions and suggestions, they would help improve our paper significantly! We are also grateful for your positive review of our work. Please let us know if you have any followup questions or if we can further clarify something.
> > >
> > > Thanks,
> > >
> > > Authors

---

### Official Review · Reviewer_WJpk · 2025-03-13

**Overall Recommendation:** 5

**Summary:**

This paper presents Paprika, a finetuning method that enables models to perform in-context rl for unknown environments. Different aspects of Paprika have been studied under different settings but not for this particular setting e.g multi-turn, not interacting with a human and more general environments. A training dataset is constructed using different LLMs for  text-based tasks. The diverse tasks have  with different  types of environmental feedback. There are two important aspects of training/fine-tuning. First the model is trained using a combined DPO and SFT loss. The second which is unique to this work is the scalable online curriculum learning to actually construct the training dataset. The coefficient of variation is used to measure the learning potential of a particular and a multi-arm bandit algorithm is used to actually make the task selection (for the training dataset). Experiments cover several aspects with the main focus on if training on n-1 tasks can lead to good performance using RL ICL on the unseen nth task.

**Claims And Evidence:**

The main idea/claim of the paper is that LLMs can be trained to perform ICL RL through the finetuning method presented (Paprika). There is not really an ablation (on the different parts of the method) so it is hard to tell which parts are most important but the experimental results do show the effectiveness of Paprika as a finetuning method.

**Essential References Not Discussed:**

Not that I know of

**Experimental Designs Or Analyses:**

The experimental design for both the task specific setting (training for ICL RL) and general setting are sound and valid. One weakness which is mentioned by the authors in the conclusion is that a lot of human effort is required in making the dataset so generalizability could be an issue.

**Methods And Evaluation Criteria:**

Given the particular task setup, the evaluation criteria makes sense. Also standard evaluation criteria are used in Section 4.1 which allow for comparisons to other methods.

**Other Comments Or Suggestions:**

N/A

**Other Strengths And Weaknesses:**

Strengths:
- Task setup: The task setup fills a hole in current RL agent work and is likely an important future area. It broadens the potential applications of agents to more real-world like scenarios rather than toy tasks.
- Generalizability: The Leave-One-Out results are particularly impressive given how different the tasks are.
- Efficiency: Showing that curriculum learning improves efficiency is also important given the high cost of trajectory generation.
Weaknesses:
- It is hard to measure different aspects of the method in part due to the exponential search space. For example, is there a way to measure how ‘correct’ the selected curriculum is? Maybe some qualitative examples to get an idea of the curriculum would be helpful.
- This is noted by the authors and above but the main weakness is the amount of human expertise involved in dataset generation. For example, humans specifically chose the tasks to make up the training set. While the tasks are diverse, it is unclear what criteria should such a training set have so that it can generalize. It could be helpful to have a short discussion on the limits of generalizability of the tasks.
Overall, this paper is a strong contribution to the overall research community. It addresses an important question and contains important empirical results that others can build on. While there are some weaknesses, the task itself is challenging and this presents a good first step towards tackling it.

**Questions For Authors:**

1. How were the particular tasks chosen and which tasks were rejected?

**Relation To Broader Scientific Literature:**

The related works section shows how Paprika fits into the broader literature. In particular, Paprika is focused on finetuning method for ICL RL generalization on multi-turn tasks. This is not a commonly used setting making it unique from other works.

**Theoretical Claims:**

N/A

---

> ### Author Rebuttal · Authors · 2025-04-01
>
> We thank the reviewer for the support and thoughtful review. We want to address their feebacks below.
>
> > There is not really an ablation (on the different parts of the method)...
>
> We have conducted additional experiments on different parts of Paprika, we list them below:
>
> 1. **Ablation on training data generation**: Since our training data is generated using high temperature (1.5) and min-p sampling with parameter 0.3, we have run an ablation to show the importance of both. On twenty questions, we see that either using a lower temperature or lower min-p sampling parameter leads to lower coverage on the training set, see [this Figure for varying Min-p at a fixed temperature of 1.5](https://ibb.co/mVsnTSsy), and [this Figure for varying temperature at a fixed Min-p parameter of 0.3](https://ibb.co/m547KZH7). The training data also affects downstream performance on held-out tasks, see [this Figure](https://ibb.co/fVYTxb1g).
>
> 2. **Ablation on fine-tuning stages**: We ran an ablation on the two different parts of our fine-tuning: namely the SFT and the RPO stage, please see [this Figure](https://ibb.co/nMtyKGXJ).
>
> If the reviewer has any other ablation in mind, we are happy to add it to the paper.
>
> > For example, is there a way to measure how ‘correct’...
>
> We thank the reviewer for this great question. In general, we found that it is hard to intuitively understand the curriculum or argue whether it’s “correct”. However, we note the following:
> 1. **Performance of the final fine-tuned model**: This is the final metric we care about. We have run our curriculum algorithm for 3 seeds and 3 rounds (with 250 tasks sampled at each round) and have an updated version of Figure 4 [here](https://ibb.co/gLn5wGPQ), it shows that the model trained on tasks selected by our curriculum outperforms that trained on uniformly selected tasks by 1.4% and 3.3% at average and pass@4 success rate respectively, demonstrating its efficacy in selecting training tasks.
> 2. **Distribution of selected tasks**: This is the other metric that is easy to understand. First, see that our defined metric for learning potential (with Llama-3.1-8B-Instruct as the policy) has an intuitive distribution over the gpt-4o-mini defined easy, medium and hard categories — easy tasks have a higher learning potential compared to medium and hard ones.
> | Category | Average Learning Potential Metric |
> |-|-|
> |Easy|0.22 |
> |Medium| 0.16|
> |Hard| 0.09|
>
> Next, notice the distribution of easy, medium and hard tasks within 20 questions: we have 477 easy, 727 medium and 296 hard questions. **Uniform sampling** respects this distribution, and samples **more medium tasks as opposed to easy tasks**. However, our curriculum algorithm follows the learning potential metric, and **samples more easy, then medium and finally the least number of hard tasks** in a sampled batch. In a batch of 250 questions, we observe the following distribution:
>
> | | Easy | Medium | Hard |
> |-|-|-|-|
> |Uniform |78 |120|52|
> |Our Curriculum|117|79|54|
>
> This shows that our curriculum has an intuitively reasonable behavior. If the reviewer has any other questions/thoughts, we would be happy to address them.
>
> > This is noted by the authors and above but the main weakness…
>
> We agree with the reviewer’s comment. To extend Paprika, one would ideally use another LLM to keep generating diverse tasks that require strategic exploration, and use the curriculum algorithm to adoptively choose which tasks to train on. We believe that this is an exciting future direction.
>
> > It could be helpful to have a short discussion on the limits of generalizability of the tasks.
>
> We thank the reviewer for this suggestion. We would edit the paper to discuss this limitation more clearly. We will emphasize the suite of tasks is diverse but it remains to be seen whether the resulting model will generalize to more temporally extended and real-world tasks.
>
> > How were the particular tasks chosen and which tasks were rejected?
>
> Some of the task groups we work with have been studied before, such as bandits [1, 2], 20 questions [3], guess my city [3] and wordle [3]. We expand upon these prior works and generate more training and test examples by prompting gpt-4o-mini with techniques from GenQA [4]. For the other tasks, we looked for tasks similar to the ones before, and GPT-4o-mini suggested battleship, minesweeper and mastermind. We came up with the cellular automata task as a toy example for iterative coding tasks with an interpreter. Finally, customer service and murder mystery are the open-ended tasks we could think of that have partial observability and fits nicely with other tasks. **We wanted as many tasks as possible, so we did not reject any tasks.**
>
> ## References
>
> [1] Can large language models explore in-context?
>
> [2] EVOLvE: Evaluating and Optimizing LLMs For Exploration
>
> [3] LMRL Gym: Benchmarks for Multi-Turn Reinforcement Learning with Language Models
>
> [4] GenQA: Generating Millions of Instructions from a Handful of Prompts

---

> > ### Comment · Reviewer_WJpk · 2025-04-02
> >
> > Thank you for the additional experiments, clarifications and detailed comments. I think this is a strong paper and keep my score at 5.

---

> > > ### Author Response · Authors · 2025-04-02
> > >
> > > Dear Reviewer WJpk,
> > >
> > > Thanks for your thoughtful feedback and suggestions, we believe they would greatly improve our paper! Please let us know if you have any followup questions or if we can further clarify something.
> > >
> > > Thanks,
> > >
> > > Authors

---

### Official Review · Reviewer_pP1H · 2025-03-22

**Overall Recommendation:** 3

**Summary:**

This paper introduces PAPRIKA, a SFT + RL approach aimed at enhancing the general decision-making and exploration capabilities of LLMs through diverse synthetic interaction data from various domains. One contribution is that it introduces 10 interesting tasks requiring interacting with the environment and reasoning about the interaction histories, such as 20 questions, etc. To improve the LLM's exploration capabilities, it generates synthetic data via rejection sampling, by either get successful trajectories or pairs of contrastive samples to do SFT, and RL. It shows the method generalizes to new domains, and also showcase some zero-shot transfer capabilities.

**Claims And Evidence:**

Most of the claims the paper made are mostly from the empirical section, on the effectiveness of the proposed method:
- PAPRIKA improves LLM decision making abiltiies. This argument is supported by Figure 2 on after applying PAPRIKA, the individual task performance is improved. One thing probably need to analyze more is that whether it shows model's better capability in solving the specific task, say the direct mapping $x\to y$, and not learning the effective method of solving $y$ given $x$. It would be great if the authors could provide more detailed statistics on the interaction trajectory? Is the model luckily committed to golden answer without effectively exploring the space first?
- PAPRIKA can teach LLMs generalizable strategies. I feel the LOO results is interesting and it does show that model's better generalization performance. One thing I am concerned is that are the "unseen" tasks are still within the same domain somehow? In these environments, are them mostly about the action is composing different questions, and the answer is yes / no? It would be great if the author could comment more on the difficulty and similarities of these tasks? I noticed Fig 3 only shows 5 domains, how about the other 5? I did not find them in Appendix as well.
- Curriculum learning can improve data efficiency of PAPRIKA. This is mostly supported by Fig 4 compared with uniform sampling, and it shows curriculum is indeed helping, compared to random sampling.

**Essential References Not Discussed:**

This would be relevant as well: LLMs Are In-Context Bandit Reinforcement Learners [https://arxiv.org/abs/2410.05362]

**Experimental Designs Or Analyses:**

See Methods And Evaluation Criteria

**Methods And Evaluation Criteria:**

- The benchmark datasets are pretty nice, and it captures more complex domains that compared to multi-arm bandit. However, as I pointed above, it would be great to have a direct comparison on these tasks, how fundamentally each task is different than the other, or is it more semantically different observation / action spaces?
- The metric used in the paper is mostly "best-of-4", why we need to report this best of $n$ performance? Can we report more metrics for better understanding? Say if we only sample once, and best of $n$ for multiple different $n$'s.
 - As I pointed before, LOO experiments in Figure 3 is missing some domains?
- The whole paper is only experimenting on the same model, llama 8b, it would be great to show the performance of PAPRIKA on multiple models, preferably different sizes, and by different organizations, might show the effectiveness of the proposed method.
- Can we list some example interaction trajectories in the Appendix for all domains? It would be great to get more detailed understanding on the complexity of the task.

**Other Comments Or Suggestions:**

See above.

**Other Strengths And Weaknesses:**

Strength: The paper is a very nice read, and it works on a very important problem, basically how to teach LLM strategic exploration and decision making capability. The proposed method is intuitive, and it shows strong generalization performance on unseen tasks.

Weakness:
- The limited experiments are only done on one specific model, might be good to include more results from diverse models.
- The evaluation metric and some missing results from what I mentioned above.
- More detailed analysis on the interaction trajectories to understand more on the improvement of the exploration capabilities.

**Questions For Authors:**

See above.

**Relation To Broader Scientific Literature:**

Related with the broader RL + LLM community, and probably agents as well.

**Theoretical Claims:**

No theoretical claims.

---

> ### Author Rebuttal · Authors · 2025-04-01
>
> We thank the reviewer for their constructive feedback, and hope to address their concerns below:
>
> > One thing probably need to analyze more is…
>
> We note that all our evaluations are conducted on held-out tasks within each group. These tasks require strategic exploration, as they are partially observable (POMDPs) and require agents to interact with the environment to gather necessary information before solving them. For instance, in Mastermind, memorizing training trajectories isn't enough --- agents must learn to make informed guesses based on prior observations. Therefore, success on held-out tasks reflects genuine improvement, not noise. Since there is no direct mapping $x \rightarrow y$ without environment interactions, higher success rates (Figure 2) and fewer turns taken (Figure 6) indicate the model is learning to explore effectively and solve tasks more strategically.
>
> > PAPRIKA can teach LLMs generalizable strategies…
>
> In our work, we use 10 different task groups (for example, 20 questions, mastermind etc.). Each task group consists of different tasks (for example, guessing ‘Apple’ and ‘Mango’ in 20 questions). Each task group also employs 2 distinct training and test subsets. For all evaluations, we use the test split, and in that sense, these tasks are **unseen**.
>
> Moreover, for the LOO experiments, we test on **completely unseen task groups**. In this way, Paprika itself tests for generalization within the same task group, and Paprika (LOO) tests for generalization to tasks that are from a different domain.
>
> Of course, the tasks have to share some “similarity” for there to be any transfer at all but we don’t think such abstract similarity makes the tasks the same domain.
>
>  > In these environments, are them mostly about the action…
>
> This is an excellent question. **The action and observation spaces of our task groups are often very different from one another.** For example, in mastermind, the agent needs to guess a 4 digit code, whereas in 20 questions it needs to ask a yes/no question about the secret topic. We discussed this in Appendix A, here we include a [summary](https://ibb.co/M5CQw5JQ).
>
> There is no obvious way to measure the difficulty of a task beyond reporting the model’s performance on each of them before fine-tuning, see Figures 2 and 5 (although this would be a great research question). Similarly, it is hard to say definitively whether one task is semantically similar to another.
>
> > I noticed Fig 3 only shows 5 domains…
>
> We have performed LOO experiments in all 10 domains. **Paprika (LOO) outperforms the instruct model in 9/10 domains**, showing remarkable generalization.
>
> 1. [This Figure](https://ibb.co/xtnt563Z) shows the performance of Paprika (LOO) with pass@4 success rate (with additional changes made to Bandits according to reviewer WvQu’s suggestion)
>
> 2. [This Figure](https://ibb.co/CfWZ65x) shows the average success rate.
>
> > The metric used in the paper is mostly "best-of-4", why we need to report this best of n performance?
>
> We used best-of-4 to account for stochasticity in environment dynamics. However, we agree that one-shot performance is also important. We have collected the following:
>
> 1. [Average success rate](https://ibb.co/0yhHNM2p)
>
> 2. [Pass@k for k = 1, 2, 3, 4, at temperature 0.7](https://ibb.co/Mk9pFtXp)
>
> > The whole paper is only experimenting on the same model…
>
> This is a good point. We have evaluated two different base models with comparable parameter count and see that they perform worse compared to Llama-3.1-8B-Instruct: [Figure](https://ibb.co/xtsyS7TG).
>
> Additionally, we have run Paprika fine-tuning on the recent Gemma-3-12B-IT, which is larger with 12B parameters. Paprika generally improves performance, and the improvement is often larger compared to that on Llama-3.1-8B-Instruct. Additionally, **Paprika fine-tuned Gemma-3-12B-IT reaches comparable or better performance to GPT-4o-mini on 7 out of 10 task groups**. Below are the most notable results (pass@4 success rate, temp 0.7, min-p 0.3):
>
> || 20 questions|Guess my city|Wordle|Battleship|
> |-|-|-|-|-|
> |Gemma-3-12 B-IT|0.60|0.75|0.33|0.40|
> |+ Paprika|0.68|0.87|0.50|0.52|
> |GPT-4o-mini|0.76|0.88|0.56|0.46|
>
> Full results: [Average](https://ibb.co/0yhHNM2p), [Pass@4](https://ibb.co/dsv8xG3q)
>
> > More detailed analysis on the interaction trajectories...
>
> We show comparison between Llama-3.1-8B-Instruct and Paprika on 3 examples:
>
> 1. 20 questions, with the secret topic being ‘prime numbers’: [Figure](https://ibb.co/N6dtBqKs)
>
> 2. 20 questions, with the secret topic being ‘Orca’: [Figure](https://ibb.co/n8KzS18y)
>
> 3. Wordle, with the secret word being ‘toast’: [Figure](https://ibb.co/G4P9gpZg)
>
> Paprika performs better information gathering and higher quality actions compared to the instruct model. We will add these and more example trajectories in the next revision of the paper.
>
> > This would be relevant as well...
>
> Thank you for the reference! We will cite and discuss this.

---

### Official Review · Reviewer_Udnq · 2025-03-24

**Overall Recommendation:** 4

**Summary:**

The paper proposes PAPRIKA, a method designed to enable large language models (LLMs) to acquire generalizable sequential decision-making capabilities via fine-tuning on synthetic interaction data. The gem of PAPRIKA lies at the use of a scalable online curriculum learning method (Sec. 3.4), where the performance variance of the current policy over the sampled tasks serves as the metric reflecting their difficulty, and the methodology is instantiated by a batched UCB algorithm. Experimental results demonstrate the effectiveness of PAPRIKA in multiple multi-turn decision-making scenarios, especially in zero-shot generalization to unseen tasks with the leave-one-out experimental design.

**Claims And Evidence:**

The three research questions raised at the beginning of Section 4 are addressed adequately in Sections 4 and Figures 2-4.

**Essential References Not Discussed:**

In Section 2, the authors stated "for most tasks, there is no known algorithm like UCB to generate good synthetic trajectories from". In fact, synthesizing trajectories for language agents has been studied in [1][2]. I suggest the authors to discuss the relationship of PAPRIKA with these prior work. For instance, could they be combined to boost the performance even further?

[1] BAGEL: Bootstrapping Agents by Guiding Exploration with Language. ICML 2024.
[2] ReAct Meets ActRe: Autonomous Annotation of Agent Trajectories for Contrastive Self-Training. COLM 2024.

**Experimental Designs Or Analyses:**

I like the design of the leave-one-out experiments. However, in Figure 3, it is shown that the performance of PAPRIKA (LOO) on Mastermind is quiet inferior to that of PAPRIKA. A straightforward explanation could be that the Mastermind task is more difficult than other tasks. But I suggest the authors to derive deeper analysis in this regard with the learning potential defined in Equation (4).

**Methods And Evaluation Criteria:**

The methods and the evaluation criterias are sound and make sense.

**Other Comments Or Suggestions:**

NA

**Other Strengths And Weaknesses:**

Strengths:

The written is clear. The experimental design and results are strong.

Weaknesses:

- The term "Curious" in the title should be clarified more clearly in the introduction section.

- In the Line 6 of Algorithm 1, \tau tasks and C samples are sampled. Is this process time-consuming? When sweeping them over several different configuations, what's their effect on sketching the difficulty of different tasks, and eventually the performance of the agents?

**Questions For Authors:**

In Section 3.2, the authors mentioned that the top-p sampling contributes to the generation of diverse yet coherent trajectories. Have the authors conducted ablation studies on this?

**Relation To Broader Scientific Literature:**

This paper is related to language agents and reinforcement learning. The main contribution from my perspective is to use the performance variance of the current policy over the sampled tasks as the metric reflecting their difficulty. Sketching the task difficulty with such a metric gives rise to the integration of curriculum learning algorithms.

**Theoretical Claims:**

The primary focus of this paper is empirical efficacy.

---

> ### Author Rebuttal · Authors · 2025-04-01
>
> We thank the reviewer for their thoughtful review and suggestions. They will greatly improve our paper!
>
> > But I suggest the authors to derive...
>
> Indeed, mastermind is the hardest task, as demonstrated by the untrained model only achieving ~4% pass@4 success rate on it, the lowest among all 10 task groups (Figure 2).
>
> It is conceptually unclear if the learning potential and LOO generalization should be related, since generalization depends on how transferable the decision making ability from training task groups is to the test task group, rather than the learning potential of the training or test task groups (which is only important for choosing which tasks to train on next). Even a highly learnable test task may not see much generalization if the training tasks do not teach any strategies that transfer to it. We are happy to incorporate any analysis if the reviewer has any suggestions.
>
> > In Section 2, the authors stated "for most tasks...
>
> This is a good point and we thank the reviewer for the references which we will discuss in the final version of the paper.
> What we meant is that we don’t always know how to generate near optimal solutions. In fact, the methods the reviewer mentions are also ways to overcome the fact that we do not know how to solve most tasks like we know how to solve bandits.
>
> We believe what sets paprika apart is that existing works focus on training to solve specific domains like web navigation whereas we are interested in whether the agents can generalize to a wide range of tasks. The motivation is that training for specific domains will not scale to all possible tasks in the world. What we actually want is an agent that is capable of general decision making so it can solve new tasks efficiently.
>
> As the reviewer pointed out, these two perspectives are complementary. These methods could be used as subroutines in Paprika to help gather good experience, and Paprika could make task-specific training more sample efficient because the models are more capable of general decision making.
>
> > The term "Curious"...
>
> We thank the reviewer for their suggestion. We would update the paper in the next iteration to clarify the terminology clearly in our introduction.
>
> We want to also briefly discuss it here: the concept of curiosity has been used in many different machine learning contexts. A popular notion is **intrinsic motivation**, where the agent is driven by an exploration bonus that is not necessarily related to the task to be achieved [1, 2]. Many works build on this notion to handle problems with sparse reward or no reward at all [3, 4]. The curiosity in this work differs from intrinsic motivation in that we focus on gathering only the information required to solve a given task rather than all the knowable information. This is closer in spirit to the original exploration-exploitation trade-off in reinforcement learning [5]. The goal is to explore to the extent that the problem can be solved but not over-explore at the cost of efficiency, by training on a wide range of problems. This can be thought of as a form of **amortized exploration**.
>
> > In the Line 6 of Algorithm 1, $\tau$ tasks and $C$ samples...
>
> Sampling trajectory can be expensive but not avoidable. It is the same as gathering experience in other RL algorithms. Since the sampling is expensive, we have only tried one configuration, the largest one our hardware can support. Namely, we sample task $\tau$ uniformly at random from task group $k^*$, and fix $C = 20$.
>
> We fully agree that different configurations can affect the estimated difficulty of a task, since the empirical estimate of the learning potential (Appendix E) might become worse if $C$ is significantly reduced, and therefore affect the final agent’s performance.
>
> > In Section 3.2, the authors mentioned that the top-p sampling...
>
> To clarify, we use min-p sampling instead of top-p, since it is known to work better at higher sampling temperatures [6]. **We have run an ablation to show the importance of min-p sampling**. On twenty questions, we see that either using a lower temperature or lower min-p parameter leads to lower coverage on the training set, as measured by pass@20 success rate, please see [this Figure for varying Min-p at a fixed temperature of 1.5](https://ibb.co/mVsnTSsy), and [this Figure for varying temperature at a fixed Min-p parameter of 0.3](https://ibb.co/m547KZH7).
>
> The training data generated by different min-p parameters and temperatures also affects downstream performance of the fine-tuned model on held-out tasks, see [this Figure](https://ibb.co/fVYTxb1g).
>
>
> # References
>
> [1] Curious model-building control systems.
>
> [2] Godel machines: Fully self-referential optimal universal self-improvers.
>
> [3] Curiosity-driven exploration by self-supervised prediction.
>
> [4] Exploration by random network distillation.
>
> [5] Reinforcement learning: An introduction
>
> [6] Turning up the heat: Min-p sampling for creative and coherent llm outputs

---

> > ### Comment · Reviewer_Udnq · 2025-04-03
> >
> > Thank you for the response. I appreciate the additional clarification and ablation studies, and I'm raising my score to 4.

---

> > > ### Author Response · Authors · 2025-04-03
> > >
> > > Dear Reviewer Udnq,
> > >
> > > We thank you for your astute observations and suggestions, they would greatly improve our work! We also thank you for the positive review of our paper. If you have any followup questions or if we can clarify something, please let us know and we would love to do so.
> > >
> > > Thanks,
> > >
> > > Authors

---

### Decision · Program_Chairs · 2025-05-01

**Decision:**

Accept (oral)

**Comment:**

In this work, the authors propose Paprika, a fine-tuning approach that is designed to teach LLMs to seek information to solve sequential decision-making problems. The core contributions include:
- A dataset of ten text-based games, each game has thousands of task variants. The games require LLM-based systems to intereact with the games in a multi-turn manner to collect necessary information to win.
- On top of the SFT + DPO model training, the authors propose a unique scalable online curriculum learning strategy. In which, the coefficient of variation is used to measure the learning potential; a multi-arm bandit algorithm is used to select training tasks.

The reviewers gave high initial scores (all positive, with an average of 3.5), the concerns are mostly clarification questions. The authors did a great job to address the concerns. In fact, the authors have successfully convinced two reviewers to further increase their scores, resulting an average score of 4. Some of the salient discussion points are:
1. The notion of `curiosity`. As the authors acknowledge in their rebuttal, their notion of curiosity is different from many RL literature, especially those of `curiosity-driven exploration` and `intrinsic motivation`, where the exploration policies' goal is to discover as much as possible the map or the game dynamics, even when the discovery may not have direct contribution to solving the task. Here, the exploration is a `amortized` version (citing the authors' word). My understanding is that the agents are required to perform an info-seeking behavior to solve the tasks because memorizing past success trajectories won't work, the `curiosity` lies in such info-seeking behavior. I encourage the authors to discuss in their camera-ready, what exactly do they mean by a `curious agent`. It is weird that the word `curious` only appeared twice in the paper, once in the title, once in the conclusion.
2. The authors added additional experimental results, including metrics other than "best-of-4"; experiments using other LLM models; additional ablation studies on training data generation and fine-tuning stages; additonal evidence on the effectiveness of the curriculum; and additional qualitative analysis.

Additional AC comments:
While I agree the 10 text-based games designed in this work are valuable, please note that text-based game / text-adventure game is an active research area. For example [1,2,3,4] are human-authored and synthetically genearted games that also satisfy the authors' criteria:
1. they are purely text based;
2. they require multi-turn interaction;
3. they are partially observable;
4. they require diverse strategies to win.

I refer the authors to [5], a good survey paper summarizes existing text-based games research effort, as a starting point to discover this area. The more I read this submission, the more I feel existing text-based game suites could greatly enrich Paprika's info-seeking capabilities.

Overall this is a strong paper with clear contributions, all reviewers are excited to see this work targeting generalizability of LLM-based agents on sequential decision making problems, and the curriculum learning method bringing efficiency to such learning procedure. I recommend to accept.

References

- [1] Interactive fiction games: A colossal adventure. Hausknecht et al., 2020.
- [2] Textworld: A learning environment for text-based games. Côté et al., 2018.
- [3] ScienceWorld: Is your agent smarter than a 5th grader? Wang et al., 2022.
- [4] Discoveryworld: A virtual environment for developing and evaluating automated scientific discovery agents. Jansen et al., 2024.
- [5]  A systematic survey of text worlds as embodied natural language environments. Jansen, 2021.